# Diverting glial glycolytic flux towards neurons is a memory-relevant role of *Drosophila* CRH-like signalling

Raquel Francés [ORCID], Yasmine Rabah, Thomas Preat [ORCID] ✉ & Pierre-Yves Plaçais [ORCID] ✉

An essential role of glial cells is to comply with the large and fluctuating energy needs of neurons. Metabolic adaptation is integral to the acute stress response, suggesting that glial cells could be major, yet overlooked, targets of stress hormones. Here we show that Dh44 neuropeptide, *Drosophila* homologue of mammalian corticotropin-releasing hormone (CRH), acts as an experience-dependent metabolic switch for glycolytic output in glia. Dh44 released by dopamine neurons limits glial fatty acid synthesis and build-up of lipid stores. Although basally active, this hormonal axis is acutely stimulated following learning of a danger-predictive cue. This results in transient suppression of glial anabolic use of pyruvate, sparing it for memory-relevant energy supply to neurons. Diverting pyruvate destination may dampen the need to upregulate glial glycolysis in response to increased neuronal demand. Although beneficial for the energy efficiency of memory formation, this mechanism reveals an ongoing competition between neuronal fuelling and glial anabolism.

The state-dependent systemic regulation of metabolism involves hormonal signalling to maintain glucose homeostasis, depending on the body's energy status and caloric intake. The keystone of this metabolic architecture is the antagonistic action of insulin and glucagon, which both induce specific responses in target cells to promote anabolism or catabolism, respectively, ultimately to maintain blood glucose levels. For example, when circulating glucose levels are high, insulin signalling executes a coordinated, full anabolic programme ranging from glucose uptake to glycolysis to lipogenesis, leading to storage of glucose carbons in fatty acid chains. The brain however has a specific metabolic status. In addition to state-dependent systemic regulation, brain tissues face an activity-dependent constraint, calling for distinct regulatory mechanisms. Estimates of the energy budget upon neuronal activation indeed conclude that neuronal electrochemical and synaptic activity represent the largest energy sink[1–3], in particular due to the Na/K ATPase pump that restores ion gradients following neuronal depolarisation (ref. 4. and references therein). Consequently, the energy needs of neurons can vary locally over several orders of magnitude in an experience-dependent and

unpredictable manner, while spanning time scales that can range from the subsecond domain (e.g., neuronal activation upon sensory stimulation) to minutes or hours (e.g., memory encoding or consolidation)[4].

While neurons are mainly responsible for this complex energy equation, its resolution appears to depend largely on glial cells. Glial cells fuel the catabolic activity of neuronal mitochondria as well as the anabolic build-up of brain energy stores, in the form of glycogen and lipid droplets (LD), of which neurons are (paradoxically) mostly devoid. Although brain energy metabolism globally relies on glucose as the primary energy source, it is well established that glucose oxidation can be sequentially split between glial cells and neurons. In mammals, glucose is mostly imported by astrocytes for partial oxidation to pyruvate through glycolysis, while pyruvate mitochondrial oxidation occurs in neurons, giving rise to a tight metabolic coupling between the two cell types sustained by metabolite transfer in the form of a lactate shuttle[5–7]. This astrocyte-neuron lactate shuttle model is proposed to underlie activity-dependent fuelling of glutamatergic synapses[5], and was recently shown to be relevant for at least some forms of synaptic plasticity[8]. In non-mammalian species as well, studies

Energy & Memory, Brain Plasticity (UMR 8249), CNRS, ESPCI Paris, PSL Research University, Paris, France. ✉e-mail: thomas.preat@espci.fr; pierre-yves.placais@espci.fr

performed in fruit flies indicate that lactate derived from glial glycolysis is also important for sustaining neuronal activity[9] and integrity[10]. Importantly, in vivo studies from various vertebrate or invertebrate species[11–13] have indicated that memory formation consistently relies on compartmentalised glucose oxidation between glycolytic glial cells and neurons that perform oxidative phosphorylation of pyruvate. We recently conducted a fine spatiotemporal dissection of this mechanism in *Drosophila*[13], revealing that memory formation following associative aversive conditioning (odour/electric shock pairing) requires increased mitochondrial pyruvate consumption in neurons of the mushroom body (MB), the Kenyon cells (KCs). This metabolic boost lasts approximately 2 h[14] and is sustained by alanine transfer from cortex glia, a perisomatic type of glial cell that closely enwraps neuronal cell bodies, where it is produced out of glycolysis-derived pyruvate[13].

Glucose can also sustain glial anabolic needs. Glycogen synthesis represents one mode of energy storage in the brain, although glycogen stores are quickly depleted upon glucose shortage[4]. Another anabolic pathway sustained by glucose carbons is de novo fatty acid synthesis, a critical step of this process being that acetyl-coA derived from glycolysis is diverted away from mitochondrial catabolism by the enzyme acetyl-coA carboxylase (ACC). Indeed, mammalian astrocytes, as well as invertebrate glial cells (including *Drosophila* cortex glia[15,16]), store LDs. LDs represent a reservoir of fatty acids with the triple role of fuelling neurons with ketone bodies in times of prolonged starvation[15,17], sustaining mitochondrial β-oxidation and resulting reactive oxygen species (ROS) production[18], and embedding per-oxidated lipids produced by neurons to avoid cellular damage[19–21]. As increasingly more roles of lipid metabolism in glia are unveiled, fatty acid synthesis appears to be another critical mode of glucose usage by glial cells.

Therefore, the metabolic architecture of brain tissue imposes on glial cells the unique and specific need for a dynamic, experience-dependent adjustment of two competing fates for pyruvate, i.e., routing towards export for catabolic usage by neurons or glial anabolic fatty acid synthesis. However, the cellular and molecular mechanisms that organise this competition depending on ongoing circuit activity or cognitive processes are unknown.

Taking advantage of the well-described metabolic regulation that occurs in *Drosophila* cortex glia upon memory formation following a stressful experience[13], we focused on the role of Dh44 peptide, the *Drosophila* ortholog of the major stress response hormone CRH[22]. We demonstrated the role of Dh44 in the metabolic tuning of cortex glia to remember a threatening situation. We showed that Dh44 peptide, released by a single pair of brain dopamine neurons, mediates a baseline limitation of de novo fatty acid synthesis in cortex glia. In addition to its basal activity, this neuropeptide signalling axis was transiently stimulated by aversive olfactory learning, thereby decreasing the anabolic consumption of glycolysis-derived pyruvate. Accordingly, disrupting Dh44 signalling impaired memory formation, which was attributable to a failure of glia-derived pyruvate supply to MB neurons, but the dual inhibition of Dh44 signalling and of ACC in cortex glia restored neuronal pyruvate consumption and memory formation. By sparing glial pyruvate for neuronal fuelling through the inhibition of a competing anabolic pathway, thereby limiting the need for increased glial glycolytic flux, acute Dh44 signalling by learning-activated neurons on glia likely permits energy-efficient memory formation.

## Results

### Dh44 signalling inhibits glial fatty acid synthesis

The reports that stress can regulate the LD content of glia[23] prompted us to investigate the effect of Dh44 signalling on glial LD. Two receptors for the Dh44 peptide have been characterised in *Drosophila*: Dh44-R1 and Dh44-R2[24]. To assess the putative metabolic effect of

Dh44 signalling on cortex glia cells, we inducibly knocked down each receptor by expressing RNAi constructs against either *Dh44-R1* or *Dh44-R2* in cortex glia using the GAL4/UAS binary expression system. RNAi expression was restrained to adulthood via the TARGET system[25]. This strategy, which relies on the ubiquitous expression of a thermosensitive GAL4 inhibitor (GAL80ts), allows inducing GAL4-mediated transgene expression acutely, rather than constitutively, by placing adult flies at an elevated temperature (30 °C), and was previously used in multiple studies from our group[13,15]. We measured the lipid droplet (LD) content in cortex glia using fluorescent staining by BODIPY, observed in the posterior cortex region of the brain[15] (Supplementary Fig. 1a). Expression of an RNAi targeting *Dh44-R1* in adult cortex glia resulted in a strong increase in LD content (Fig. 1a). Targeting the Dh44-R2 receptor, in contrast, produced no detectable change (Fig. 1a), suggestive of a specific effect of Dh44-R1 activation on LD content. Notably, the efficacy of both RNAi constructs was assessed by RT-qPCR in previous studies (~45% decrease in *Dh44-R1* mRNA with pan-neuronal expression of *Dh44-R1* RNAi[26,27]; ~60% decrease in *Dh44-R2* mRNA with whole-body expression of the *Dh44-R2* RNAi[27]). Expression of the *Dh44-R1* RNAi in cortex glia yielded a moderate but significant 10–15% decrease in total head *Dh44-R1* mRNA (Supplementary Fig. 1b), in support of this receptor being expressed in cortex glia, although the small decrease we measured is consistent with this receptor being mostly expressed in neurons.

As LD formation is a cellular mechanism for storing excess fatty acid[19], we reasoned that Dh44-R1 receptor activity may constitutively limit fatty acid synthesis in glia to levels that induce moderate LD formation. ACC is the key enzyme in the pathway of de novo fatty acid synthesis, which recruits carbon from acetyl-coA molecules into fatty acid carbon chains. Although acute knockdown of *Dh44-R1* alone in cortex glia caused an increased LD content, co-expressing two RNAis against both *Dh44-R1* and *ACC* in cortex glia failed to induce such a fatty glia phenotype (Fig. 1b). Although this result is consistent with *Dh44-R1* inhibition increasing fatty acid production, an alternative explanation could be that *Dh44-R1* inhibition decreases fatty acid usage. In this case, LD increase upon *Dh44-R1* knockdown could result from the accumulation of unconsumed fatty acid derived from ACC activity. However, expressing *ACC* RNAi alone did not in itself induce a decrease in the basal LD content (Supplementary Fig. 1c), which shows that the contribution of ACC activity to LD content in cortex glia of wild type flies is negligible. We thus conclude that Dh44-R1 signalling in cortex glia is a mechanism of constitutive inhibition of ACC-mediated de novo fatty acid synthesis.

### A single pair of Dh44-expressing dopamine neurons inhibits fatty acid synthesis in cortex glia

Since our results point to an ongoing inhibition of fatty acid synthesis through Dh44-R1 activation, we next searched for the secreting source of the Dh44 peptide that would cause this effect. The canonical source of this peptide is a group of six cells located within the cluster of neuroendocrine cells in the brain area called the pars intercerebralis (PI) (ref. 28. Supplementary Fig. 2a). Using an RNAi construct targeted against *Dh44*, the efficacy of which was previously verified[29], we knocked down *Dh44* expression in Dh44 PI neurons in adult flies. Surprisingly, this had no effect on the LD content in cortex glia (Supplementary Fig. 2b), suggestive of an alternative source of Dh44 peptide for modulating LD synthesis in glia.

A recent study[30] presenting single-cell transcriptomics of input and output neurons of the MB fortuitously reported high levels of *Dh44* mRNA in a specific pair of neurons (MP1 neurons, or PPL1-γ1>pedc, Supplementary Fig. 2a), which were previously characterised as MB-afferent dopaminergic neurons[31–34] with multiple roles in learning and memory processes[33,35–38]. Using a previously published antibody against Dh44[39], we confirmed that a high level of Dh44 peptide could be detected in the cell bodies of MP1 neurons, which was

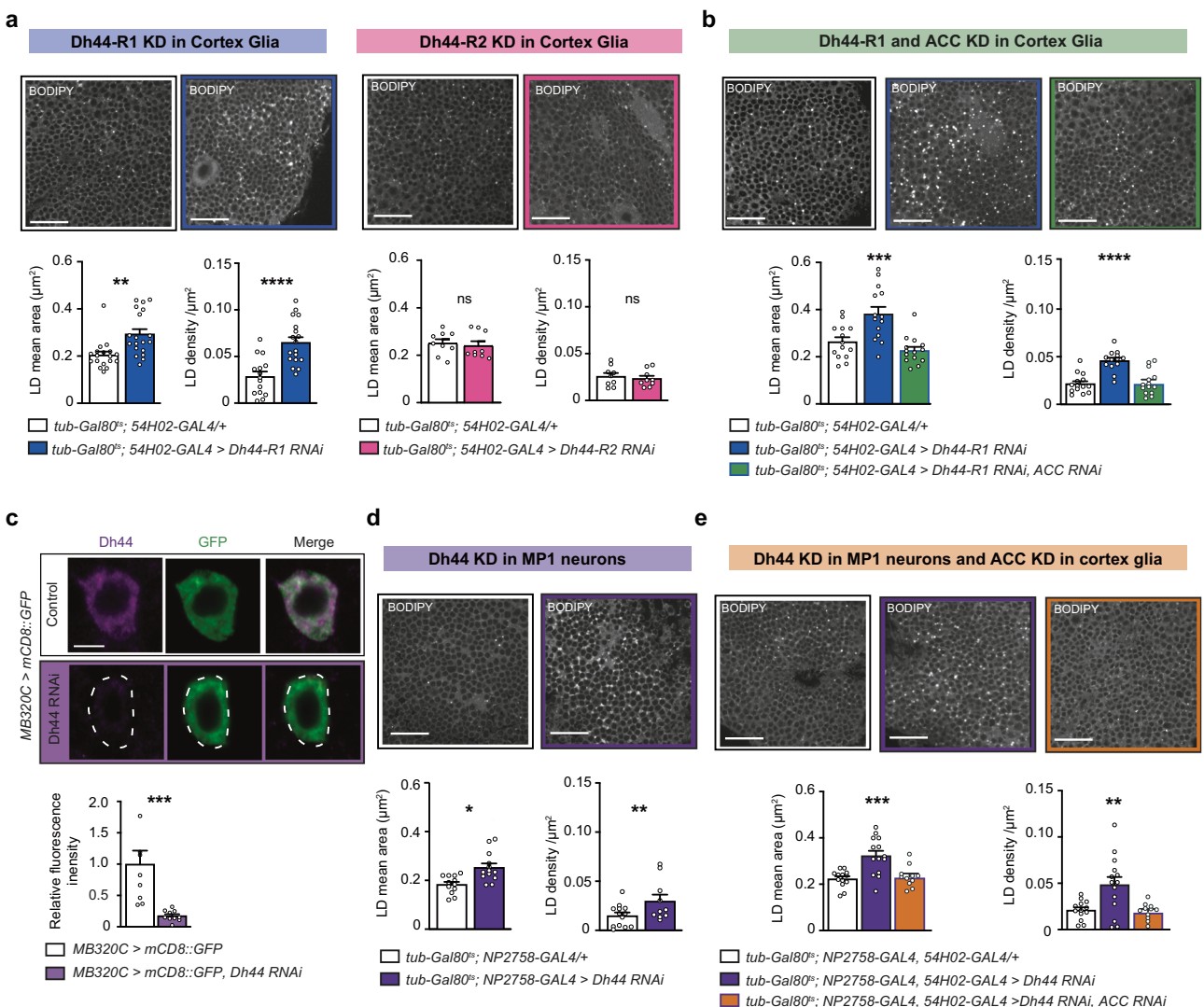

**Fig. 1 | An MP1 neuron to cortex glia Dh44 signalling axis inhibits glial fatty acid synthesis. a** BODIPY staining in the MB cortex region and quantification of LD mean area and density comparing genotypic control flies to flies expressing in adult cortex glia the *Dh44-R1* RNAi ($n = 18$; mean area: $t_{35} = 3.44$ $p = 0.0015$; density: $t_{35} = 3.44$ $p = 0.0015$) or the *Dh44-R2* RNAi ($n = 9$; mean area: $t_{16} = 0.43$, $p = 0.67$; density: $n = 9$, $t_{16} = 0.52$, $p = 0.60$). **b** BODIPY staining in the MB cortex region and quantification comparing flies expressing the *Dh44-R1* RNAi alone, or *Dh44-R1* RNAi and *ACC* RNAi together in adult cortex glia with genotypic control ($n = 14$; mean area: $F_{2,39} = 12.88$, $p = 5 \times 10^{-5}$; density: $F_{2,38} = 19.20$, $p = 2 \times 10^{-6}$). **c** The specific driver line MB320C allowed mCD8::GFP expression in MP1 neurons. Immunostaining against GFP (green) and Dh44 (magenta) revealed expression of Dh44 in MP1 neurons (merge), strongly reduced by co-expression of *Dh44* RNAi ($n = 9$, $t_{17} = 3.96$,

$p = 0.001$). **d** BODIPY staining in the MB cortex region and quantification comparing flies expressing the *Dh44* RNAi in adult MP1 neurons with genotypic control ($n = 13$; mean area: $t_{25} = 3.08$, $p = 0.005$; density: $n = 13$, $t_{21} = 2.16$, $p = 0.04$). **e** BODIPY staining in the MB cortex region and quantification comparing flies expressing the *Dh44* RNAi alone or both *Dh44* RNAi and *ACC* RNAi, in adult MP1 neurons and cortex glia, with genotypic control ($n = 14$; mean area: $F_{2,35} = 11.66$, $p = 0.0001$; density: $F_{2,35} = 8.12$, $p = 0.0013$). RNAi lines KK108591 (*Dh44-R1*), JF03289 (*Dh44-R2*), GD3482 (*ACC*) and JF01822 (*Dh44*) were used in this figure. Data are represented as mean ± SEM. Scale bars indicate 5 µm (**c**) or 20 µm (**a**, **b**, **d**, **e**). ns: not significant, $p > 0.05$, *$p < 0.05$, **$p < 0.01$, ****$p < 0.0001$ by two-tailed Student's $t$ test (**a**, **c**, **d**) or Tukey's pairwise comparison following one-way ANOVA (**b**, **e**). Source data are provided as a Source Data file.

strongly reduced by *Dh44* RNAi expression (Fig. 1c). In contrast with PI neurons, adult stage expression of *Dh44* RNAi in MP1 neurons reproduced the fatty glia phenotype that was observed upon *Dh44-R1* knockdown in cortex glia (Fig. 1d). We sought to test if this increase was also dependent on ACC activity in cortex glia. It was technically not possible to simultaneously express *Dh44* RNAi in MP1 neurons only and ACC RNAi in cortex glia only, as we lack two independent inducible binary expression systems usable in *Drosophila*. As an alternative strategy to achieve this goal, we expressed both RNAis in both cell types. This choice was supported by published single-cell transcriptomics datasets[30,40] indicating well-segregated expression of ACC or Dh44 in cortex glia and MP1 neurons, respectively (Supplementary Note 1), which we could confirm at the protein level: a previously

published *Drosophila* ACC antibody[41] revealed strong cytosolic staining in cortex glia, whereas cytosolic staining was absent in MP1 neurons (Supplementary Fig. 2c); conversely, Dh44 staining confirmed a much stronger expression in MP1 neurons than in cortex glia (Supplementary Fig. 2d). Following this strategy, expressing both *Dh44* RNAi and *ACC* RNAi in MP1 neurons and in cortex glia suppressed the increased LD content (Fig. 1e). As a functional control of the dual RNAi expression, we ensured that increased LD content in cortex glia was still observed when both RNAi were expressed solely in MP1 neurons (Supplementary Fig. 2e). Our results therefore reveal a neuropeptide axis linking MP1 dopamine neurons to cortex glia which, through Dh44 signalling on Dh44-R1 receptor, represses ACC-mediated fatty acid synthesis in glial cells.

## Dh44 secretion by MP1 neurons is acutely increased after olfactory learning

MP1 neurons and cortex glia were both previously reported to be involved in memory formation, using a well-established aversive associative olfactory conditioning paradigm. In particular, MP1 neurons signal electric shock perception to MB neurons during learning[35], whereas cortex glia provide metabolic support after conditioning, as described above[13]. We therefore asked if the neuropeptide signalling axis revealed here could play a role in memory using this paradigm. A single cycle of training (1x training) induces memory in the form of learned odour avoidance that lasts for several hours[42]. Memory measured 3 h after conditioning results from the parallel retrieval of middle-term memory (MTM), a labile form of memory that is erased by cold shock-induced anaesthesia as well as anaesthesia-resistant memory (ARM), a form of memory that is resistant to cold shock. The distinction between MTM and ARM is made at the circuit level, as they are encoded in distinct subsets of MB neurons and recruit distinct MB output circuits upon retrieval[43]. RNAi-mediated knockdown of *Dh44* expression in MP1 neurons led to a strong impairment of memory measured 3 h after 1x training (Fig. 2a). A cold shock treatment performed between training and memory testing indicated that the memory impairment specifically affects MTM (Fig. 2a). In the absence of the temperature shift that induces RNAi expression, no memory defect was observed (Fig. 2a), indicating that the memory defect was

not due to leaky RNAi expression during development. *Dh44* RNAi expression did not alter naïve odour and electric shock avoidance (Supplementary Table 1), indicating that the observed memory impairment was not due to impaired sensory perception. As the NP2758-GAL4 line used to drive RNAi expression does not exclusively target MP1 neurons, we adopted a control intersectional strategy involving the *TH-GAL80* transgene that specifically inhibits GAL4 expression in MP1 neurons, within the NP2758 expression pattern[44]. For this, no memory defect was observed (Fig. 2b), further supporting that Dh44 is required in MP1 neurons for memory formation. To confirm these behavioural results, we performed the same series of experiments with a second RNAi construct that targets *Dh44*, which yielded similar results (Supplementary Fig. 3a, Supplementary Table 1). This second RNAi was previously shown to be as efficient as the first one at decreasing *Dh44* mRNA levels (see RT-qPCR measurement in ref. 29). To further investigate whether acute Dh44 signalling occurred after learning, we examined the timecourse of Dh44 staining in the cell bodies of MP1 neurons after learning, assuming that increased Dh44 release by MP1 neurons would result in a depletion of Dh44 staining, as previously reported in myosuppressin-expressing neurons after mating[45]. We compared flies submitted to 1x training and flies submitted to an unpaired protocol, during which the same odorants and electric shocks were delivered, albeit separated in time so that no associative memory was formed. Immediately after training,

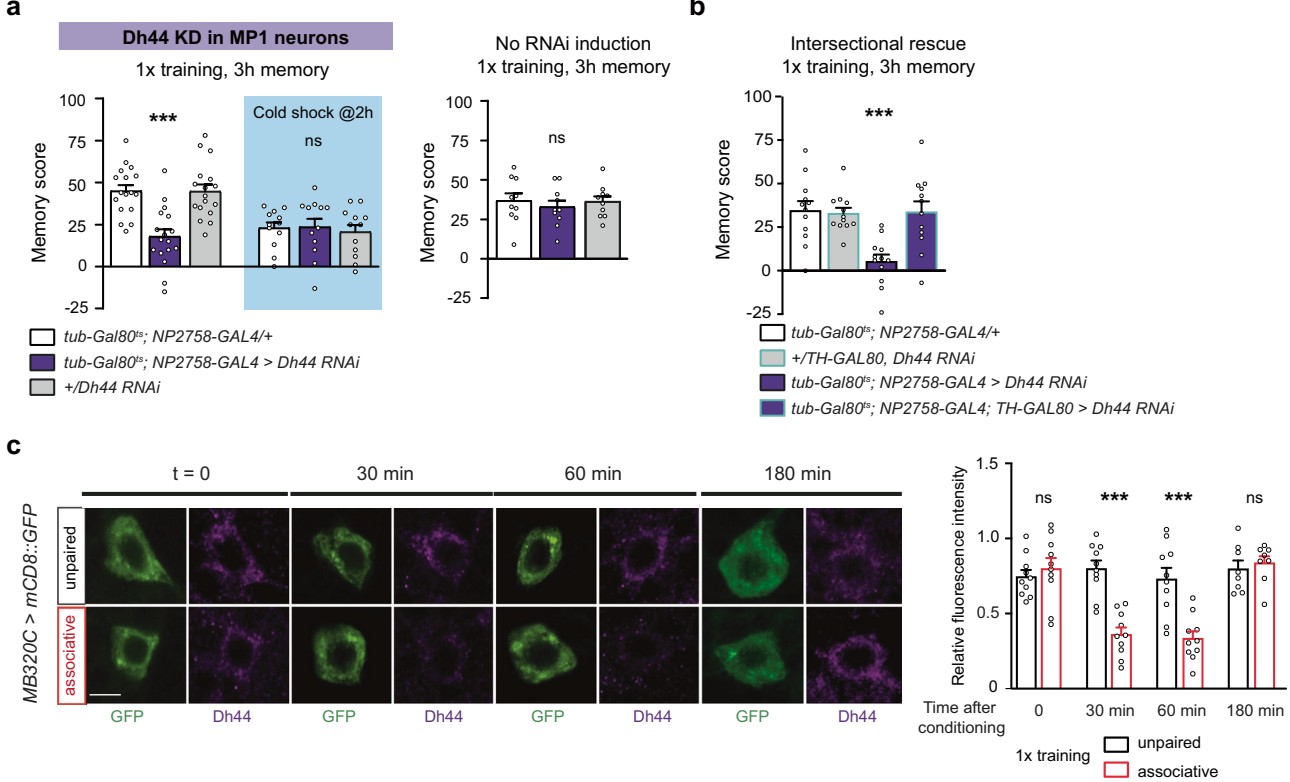

**Fig. 2 | Acute Dh44 secretion by MP1 neurons after olfactory learning is required for memory. a** *Dh44* knockdown (KD) in MP1 neurons impaired the total memory measured 3 h after single-cycle training ($n = 17$, $F_{2,48} = 15.55$, $p = 6 \times 10^{-6}$) but not cold shock-resistant memory ($n = 12$, $F_{2,33} = 0.13$, $p = 0.87$). Without induction of RNAi expression, flies showed normal memory after single-cycle training ($n = 10$, $F_{2,27} = 0.27$, $p = 0.76$). **b** The addition of the *TH-GAL80* transgene makes it possible to remove MP1 neurons from the expression pattern of the NP2758-GAL4 line. As such, expression of the *Dh44* RNAi failed to induce a memory defect ($n = 12$, $F_{3,44} = 1.70$, $p = 0.0001$). Note that the more specific MB320C split-GAL4 driver used in the MP1 staining experiments utilises a p65 activating domain[34] and thus cannot be used in an inducible manner, as it is insensitive to GAL80ts-mediated inhibition.

**c** Immunostaining series against Dh44 in GFP-labelled MP1 neurons at different time points after single-cycle associative training, or the unpaired conditioning protocol. No difference in relative fluorescence intensity between the two conditions was detected immediately after training ($n = 10$, $t_{18} = 0.77$, $p = 0.44$), whereas Dh44 fluorescence was significantly reduced at 30 min ($n = 10$, $t_{18} = 5.21$, $p = 0.00059$) and 60 min ($n = 10$, $t_{18} = 4.45$, $p = 0.0003$). Dh44 fluorescence was restored after 3 h ($n = 8$, $t_{14} = 0.53$, $p = 0.59$). Scale bar: 5 μm. RNAi line JF01822 (*Dh44*) was used in this figure. Data are represented as mean ± SEM. ns: not significant, $p > 0.05$, ***$p < 0.001$ by two-tailed Student's $t$ test (**d**) or Tukey's pairwise comparison following one-way ANOVA (**a–c**). Source data are provided as a Source Data file.

Dh44 staining was similar between the two groups of flies (Fig. 2c). In contrast, a strong decrease in Dh44 staining was observed both 30 and 60 min after 1x training as compared to the unpaired protocol. Dh44 levels returned to control levels 3 h after training (Fig. 2c). A similar measurement performed in Dh44 neurons of the PI revealed no difference in Dh44 staining between 1x-trained and unpaired flies (Supplementary Fig. 3b). These data are consistent with a post-learning acute, specific increase in Dh44 release from MP1 neurons starting within 30 min after learning and lasting 1−2 h, which explains why inhibiting Dh44 expression in MP1 neurons induces an MTM defect.

## Dh44 signalling from MP1 neurons to cortex glia is required for glia neuron metabolic coupling during memory formation

After showing that Dh44 peptide release from MP1 neurons acts on cortex glia in naïve flies, we asked if its secretion following associative learning by these neurons acts on the same target. RNAi-mediated downregulation of *Dh44-R1* expression in cortex glia at the adult stage induced an MTM defect (Fig. 3a) similar to the phenotype obtained upon *Dh44* knockdown in MP1 neurons. No defect was observed without induction of RNAi expression (Fig. 3a), and naïve odour and shock avoidance were unaffected by *Dh44-R1* RNAi expression in cortex glia (Supplementary Table 2). These results were confirmed using a second RNAi targeting *Dh44-R1* (Supplementary Fig. 4a, Supplementary Table 2), which was previously shown to efficiently downregulate *Dh44-R1* mRNA (~50% decrease in qPCR in ref. 27). Notably, knockdown of the *Dh44-R2* receptor did not impair memory performance (Fig. 3b), confirming the specificity of the Dh44-R1 receptor in mediating the Dh44 effect on memory in cortex glia, as was the case in a naïve context.

We previously described the involvement of cortex glia in providing energy to neurons for MTM formation[13]. As the metabolic upregulation in MB neurons after 1x training lasts approximately 2 h[14], matching the timecourse of Dh44 release by MP1 neurons, we hypothesised that the stimulation of the MP1 neuron-cortex glia Dh44 signalling axis may be required for pyruvate supply to MB neuronal mitochondria during memory formation. We expressed a genetically-encoded FRET pyruvate reporter (Pyronic) in MB neurons[46] and used a protocol that we had previously established[13,36] to monitor mitochondrial pyruvate consumption in MB neurons by 2-photon in vivo imaging. Briefly, this protocol consists in measuring the rate of pyruvate accumulation in MB neurons following the acute pharmacological inhibition of the mitochondrial respiratory chain using sodium azide. In a previous work, we showed that this measurement correlates with the activity level of the pyruvate dehydrogenase (PDH) complex[36]. As previously reported[13], we observed 1 h after 1x training an increased rate of pyruvate consumption in MB neurons, as compared to flies subjected to an unpaired protocol (Fig. 3c). When *Dh44-R1* was knocked down in cortex glia, we failed to observe this metabolic upregulation (Fig. 3c). A similar result was obtained upon *Dh44* knockdown in MP1 neurons (Fig. 3d). In our previous work we had shown that feeding flies with L-alanine-enriched food before training could rescue the impairment in MTM that arose from defective alanine export by cortex glia[13]. The same procedure of L-alanine feeding also rescued MTM defects due to the knockdown of *Dh44-R1* in cortex glia or of *Dh44* in MP1 neurons (Fig. 3e, f). These results altogether indicate that Dh44 release from MP1 neurons after learning activates Dh44-R1 in cortex glia to enable glia-neuron metabolic coupling, allowing MTM formation.

## Learning-induced Dh44 signalling allows a local increase of pyruvate levels in cortex glia

How does Dh44-R1 activation in cortex glia after learning enable energy transfer to MB neurons? Evaluating intracellular pyruvate levels requires accurate measurement of the steady-state FRET of the Pyronic sensor, which can be achieved by measuring the fluorescence lifetime of the donor fluorophore (FRET-FLIM)[47]. After validating that this approach could report an increase in pyruvate level in vivo in KCs (Supplementary Fig. 5a), we adopted this strategy to measure pyruvate levels in cortex glia after learning. An independent DsRed labelling of KCs allowed to distinguish cortex glia around KCs from cortex glia outside the MB region (Fig. 4a). 1 h after associative learning, we observed an increased pyruvate level, specifically in cortex glia surrounding KCs, compared to control flies (subjected to the unpaired protocol) (Fig. 4b). In those control flies, pyruvate levels in cortex glia around KCs and outside MB were strongly correlated, but this correlation was lost in flies that received associative learning protocol (Fig. 4b), indicative of a local regulation of cortex glia pyruvate level in the MB area, independently of the rest of the brain.

Knockdown of *Dh44-R1* in cortex glia did not yield significant alteration of pyruvate levels in cortex glia in naïve flies (Supplementary Fig. 5b). However, those flies failed to increase their pyruvate level following associative learning (Fig. 4c), and MB cortex glia remained strongly correlated to cortex glia outside MB in that case (Fig. 4c). These experiments further support that Dh44-R1 signalling after learning sustains an increase in glial pyruvate specific to the MB region, which is required for the energy transfer to MB neurons for memory formation.

## ACC inhibition in cortex glia rescues the memory impairment induced by defective Dh44 signalling

How does acute stimulation of Dh44-R1 signalling in cortex glia results in increased pyruvate concentration? On the one hand, we showed that one effect downstream of Dh44-R1 in naïve flies is to inhibit fatty acid synthesis mediated by ACC. Pyruvate is the main carbon source for de novo fatty acid synthesis, as it can be readily converted into acetyl-coA, the substrate of ACC, by mitochondrial PDH. On the other hand, energy supply to neurons consumes glial pyruvate for transamination into alanine, which is transferred to neurons[13]. Therefore, the two processes may compete for the same metabolite pool, i.e., glycolysis-derived pyruvate. In this case, lifting the inhibition of fatty acid synthesis through defective Dh44/Dh44-R1 signalling would result in a leak of pyruvate towards the ACC-mediated pathway, preventing the increase in pyruvate levels and the availability of this metabolite for sustaining glia-neuron metabolic coupling.

To test this hypothesis, we reasoned that inhibition of ACC may compensate the effect of impaired Dh44/Dh44-R1 signalling. We co-expressed RNAis against both *ACC* and *Dh44-R1* in the cortex glia of adult flies and tested their memory ability. While we confirmed that flies expressing the single *Dh44-R1* RNAi displayed a memory defect after 1x training, additional inhibition of ACC completely rescued memory performance (Fig. 5a). This rescue was also observed in combination with the second *Dh44-R1* RNAi that we previously used (Supplementary Fig. 6a). At the cellular level, in vivo pyruvate imaging experiments also revealed that the dual knockdown of *ACC* and *Dh44-R1* in cortex glia re-established the increased pyruvate consumption by mitochondria in MB neurons (Fig. 5b), providing a cellular explanation for the behavioural rescue of memory performance.

Finally, expressing *ACC* and *Dh44* RNAis in MP1 neurons and in cortex glia also rescued the memory defect induced by *Dh44* inhibition in MP1 neurons (Fig. 5c). As a control of this latest experiment, we verified that the expression of *ACC* and *Dh44* RNAis only in MP1 neurons could still produce an observable memory defect (Supplementary Fig. 6b), and that expression of *Dh44* RNAi only in cortex glia had no effect on memory (Supplementary Fig. 6c). In addition to further supporting that Dh44 release is necessary for MTM formation due to its inhibitory effect on fatty acid synthesis in cortex glia, this experiment also demonstrates that no other action of MP1-released Dh44 peptide, either in cortex glia or in other cell types, is relevant for MTM formation.

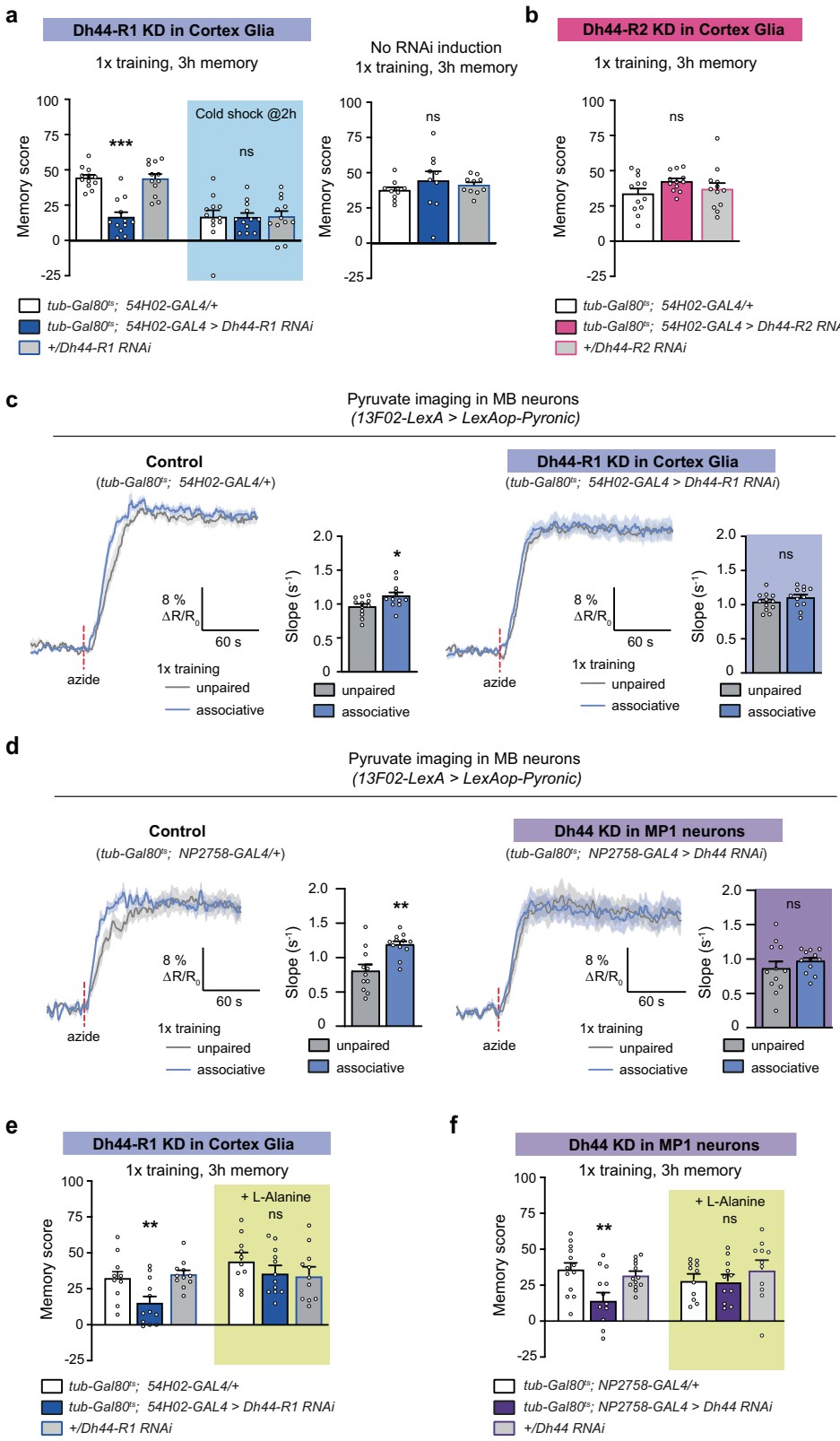

## The Dh44/Dh44-R1 signalling axis is necessary for long-term memory

When submitted to repeated odour/shock associations, flies can form more persistent consolidated memories. Indeed, repeated consecutive training cycles (massed training) induce memory that persists up to 48 h[48]. Flies, as well as many other species including humans, demonstrate a spacing effect, in which multiple training cycles spaced by rest intervals (spaced training) induce long-term memory (LTM) that can last up to one week[48]. LTM, in contrast to memory resulting from a massed training, is encoded in the same MB neurons as MTM, and its retrieval mobilises the same MB output circuit[43,49–51]. Consequently, many cellular mechanisms that are required for MTM are also necessary for LTM formation. In particular, any manipulation that impedes the transfer of alanine from cortex glia to neurons and the

**Fig. 3 | MP1 neuron to cortex glia Dh44 signalling is required for glia neuron metabolic coupling during memory formation. a** *Dh44-R1* KD in cortex glia impaired the total memory measured 3 h after single-cycle training ($n = 12$, $F_{2,33} = 1.03$, $p = 1.2 \times 10^{-6}$), but not cold shock-resistant memory ($n = 12$, $F_{2,33} = 0.39$, $p = 0.99$). Without induction of RNAi expression, flies showed normal memory after single-cycle training ($n = 10$, $F_{2,27} = 0.72$, $p = 0.49$). **b** *Dh44-R2* KD in adult cortex glia showed normal memory after single-cycle training ($n = 12$, $F_{2,27} = 1.58$, $p = 0.22$). **c** Single-cycle training elicited a faster pyruvate accumulation in MB neuron axons following azide application (5 mM) as compared to non-associative unpaired conditioning ($n = 12$, $t_{22} = 2.45$, $p = 0.022$). This effect was abolished by *Dh44-R1* KD in adult cortex glia ($n = 12$, $t_{22} = 1.11$, $p = 0.27$). **d** Similarly, *Dh44* KD in MP1 neurons impaired the single-cycle induced increase in pyruvate accumulation in MB neuron

axons following sodium azide application ($n = 12$, $t_{22} = 0.93$, $p = 0.36$; positive control: $n = 12$, $t_{21} = 2.45$, $p = 0.01$). L-Alanine feeding (60 mM) for 72 h before training rescued memory defects induced by *Dh44-R1* KD in adult cortex glia (**e** no alanine: $n = 10;12;11$ from left to right; $F_{2,30} = 1.29$, $p = 0.003$; alanine feeding, $n = 10;12;11$ from left to right, $F_{2,30} = 1.08$, $p = 0.35$) or by *Dh44* KD in adult MP1 neurons (**f** no alanine: $n = 13;12;13$ from left to right, $F_{2,36} = 5.89$, $p = 0.006$; alanine feeding, $n = 11$, $F_{2,30} = 0.81$, $p = 0.45$). RNAi lines KK108591 (*Dh44-R1*), JF03289 (*Dh44-R2*), and JF01822 (*Dh44*) were used in this figure. Data are represented as mean ± SEM. ns: not significant, $p > 0.05$, *$p < 0.05$, **$p < 0.01$, ***$p < 0.001$ by two-tailed Student's t test (**c**, **d**) or Tukey's pairwise comparison following one-way ANOVA (**a–c**, **e**, **f**). Source data are provided as a Source Data file.

---

increased pyruvate consumption in MB neurons after 1x training also resulted in LTM deficits, although memory after massed training was not affected[13]. We thus sought to assess whether the neuropeptide axis we revealed here as sustaining MTM is also involved in LTM. *Dh44* knockdown in MP1 neurons at the adult stage caused a memory defect when measured 24 h after spaced training (Fig. 6a). In the absence of RNAi induction, no effect was observed (Fig. 6a). In contrast, no effect was observed following massed training (Fig. 6a, Supplementary Fig. 7a for intersectional experiment using *TH-GAL80*). The differential effect between spaced and massed training was confirmed with the second RNAi targeting *Dh44* (Supplementary Fig. 7b). Accordingly, decreased *Dh44-R1* expression in cortex glia, using the two different RNAis, also induced a memory defect after spaced training, but not after massed training (Fig. 6b, Supplementary Fig. 7c (2nd RNAi)). Finally, LTM defects due to *Dh44-R1* knockdown in cortex glia (Fig. 6c, Supplementary Fig. 7d) or to *Dh44* knockdown in MP1 neurons (Fig. 6d; Supplementary Fig. 7e) could be rescued by additional knockdown of *ACC* in cortex glia, confirming that MTM and LTM are affected through the same mechanism.

## PKA mediates the inhibitory action of Dh44 signalling on FA synthesis

Dh44-R1 is a G-protein coupled receptor, which in other cellular systems was shown to be positively coupled to cAMP signalling[39,52]. We thus wondered whether PKA activation could mediate FA synthesis inhibition. We used an RNAi construct to knock-down *PKA-C1*, the catalytic subunit of PKA, that efficiently reduced *PKA-C1* mRNA levels when expressed in all glial cells (Supplementary Table 3). Strikingly, knockdown of *PKA-C1* in cortex glia resulted in increased LD content (Fig. 7a), which was rescued by concurrent knockdown of *ACC* (Fig. 7b), consistent with PKA acting downstream of Dh44-R1. In memory assays, knockdown of *PKA-C1* induced memory defects that were similar to those obtained in the case of *Dh44-R1* knockdown: MTM following 1x training was impaired, as well as LTM measured 24 h after spaced training; memory measured 24 h after massed training was, in contrast, unaffected (Fig. 7c, Supplementary Fig. 8a, Supplementary Table 4). These phenotypes were confirmed when using a second *PKA-C1* RNAi (Supplementary Fig. 8b). Finally, a knockdown of both *PKA-C1* and *ACC* in cortex glia rescued MTM and LTM deficits (Fig. 7d, Supplementary Fig. 8c). Overall, these results point to PKA as a mediator of Dh44-R1 inhibitory effect on FA synthesis in cortex glia.

## Discussion

In this study we uncovered a neuropeptide signalling axis in *Drosophila* involving the CRH-like Dh44 peptide. This peptide is secreted by a single pair of neurons in the brain that were previously characterised as dopamine neurons with multiple important roles in learning and memory, and acts on the Dh44-R1 receptor in cortex glia, a type of glial cell involved in metabolic support to neurons[13,15,53]. This signalling exerts a basal inhibition of ACC-mediated fatty acid synthesis in cortex glia, which is mediated by PKA, and was further activated upon associative learning. Dh44 release from MP1 neurons was acutely enhanced

after associative learning for aversive memory formation, which allowed transfer of pyruvate-derived alanine from cortex glia to neurons through inhibition of the competing fatty acid synthesis pathway (Fig. 8). Our work reveals an additional dimension to the already large repertoire of functions sustained by MP1 neurons in memory processes. We also unveil a scheme for activity-dependent metabolic regulation of glial cells to fuel memory formation in neurons, which aligns with the increasingly prevailing view that the brain functions in an energy-sparing mode even without restricted nutrient availability[54–56]. As discussed below, several lines of evidence from the literature indicate that this scheme could be conserved in mammals.

### MP1 neurons show functional versatility in memory formation and regulation

Our results demonstrate that Dh44 signalling on cortex glia has a role in inhibiting fatty acid synthesis to favour metabolite export to MB neurons. Surprisingly, the relevant source of Dh44 here was not its canonical source in the fly brain – the cluster of six Dh44-positive cells in the PI. Instead, we revealed a role for a previously unsuspected source of Dh44 in the fly brain, a single pair of neurons called MP1 (or alternatively PPL1-γ1>pedc[34]), which have been well-characterised as MB-afferent dopaminergic neurons. Single-cell transcriptomic profiling of the ~20 types of MB-afferent dopamine neurons previously revealed that among that population, MP1 neurons uniquely express high levels of Dh44 mRNA[30], which we confirmed here at the protein level in the cell bodies of MP1 neurons. As conveyed by its alternative nomenclature, the MP1 neurons send two separate branches to distinct and spatially-separated MB compartments: the γ1 compartment, where they contact the γ class of KCs, the MB intrinsic neurons; and the distal peduncle, where they contact α/β KCs. Because γ KCs support short-term memory while α/β KCs support MTM and LTM[43,57,58], MP1 neurons are well-positioned to exert multiple roles, both during and after aversive olfactory learning, in the formation of successive memory phases. A first group of MP1-related effects can be understood through a purely synaptic action in the γ1 compartment, building memory-relevant synaptic plasticity through dopamine action during learning[32,35,37,58–60], and progressively erasing it through glutamate[61] and nitric oxide[30] signalling after learning, while the interaction between glutamate and nitric oxide remains to be investigated.

In the present study we revealed another post-learning role for MP1 neurons involving neuropeptide release. Our results show that MP1 Dh44 signalling, through its metabolic action on cortex glia, is required for the MTM component of memory after 1x training and for LTM after spaced training, which both involve the α/β class of KCs[43], while being dispensable for ARM after 1x training or after massed training, which involve γ or α'/β' KCs, respectively. Strikingly, in contrast to the other post-learning roles of MP1 neurons involving glutamate and nitric oxide, Dh44 release from MP1 neurons positively modulates memory, revealing a radically opposing action of these neurons after learning on memory formed in α/β or γ KCs, respectively. Dh44 staining after learning indicated an acute release of Dh44 peptide by MP1 neurons upon associative learning (coincident odour

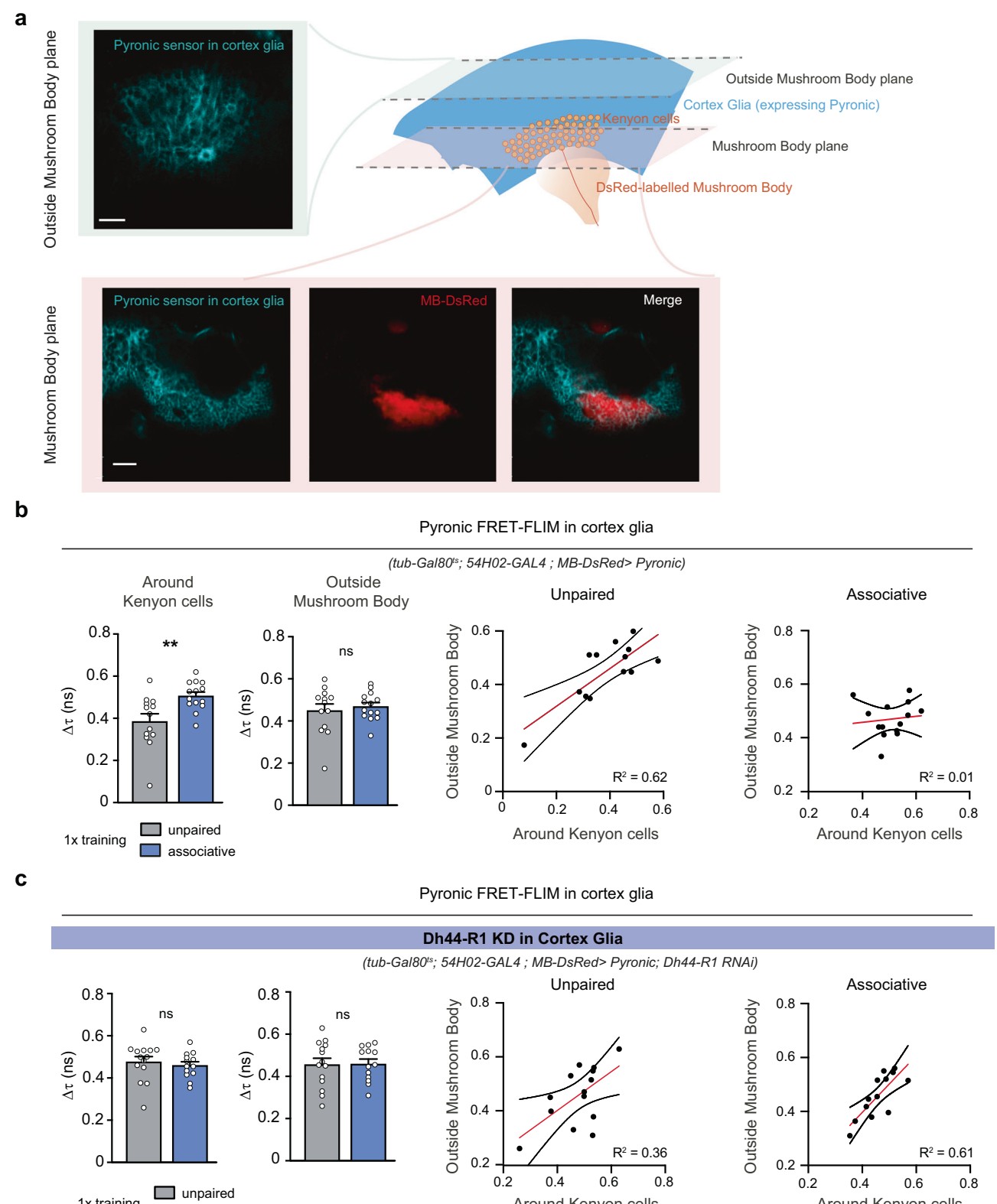

and electric shocks delivery) but not in the unpaired condition where shocks and odours are not associated. How does coincidence information reaches MP1 neurons? These neurons reportedly respond strongly to aversive stimuli such as electric shocks, but also to odour perception, although more moderately[62,63]. It is thus conceivable that some coincidence detection mechanism may exist within these neurons. Alternatively, coincidence information could directly come from

KCs, where coincidence detection is well-documented, as MP1 neurons are strongly connected post-synaptically to γ and α/β KCs[64–68]. Finally, the differential peptide secretion between associative and unpaired conditionings could be regulated by ensheathing glia, a neuropil-associated glia distinct from cortex glia: it was indeed recently shown that glutamatergic transmission from ensheathing glia on a population of dopamine neurons that include MP1 neurons is required to prevent

**Fig. 4 | FRET-FLIM measurement of pyruvate levels in cortex glia after learning. a** FRET-FLIM acquisitions were performed on flies expressing Pyronic sensor in cortex glia (*tub-GAL80*^ts; *54H02-GAL4 > UAS-Pyronic*) and DsRed in MB neurons (*MB-DsRed*). Fluorescence lifetime of mTFP (donor fluorophore of Pyronic) and fluorescence intensity of DsRed were recorded in two different planes, one including KCs somatas, and one above the MB region (along the dorso-ventral axis), as indicated by the representative images shown (scale bars: 20 μm; 54 flies were assayed in total). **b** An increase in mTFP lifetime (i.e., a decrease in FRET) corresponds to an increase in pyruvate level. Measurements were performed 1 h after training. Associative learning induced an increase in pyruvate levels in cortex glia surrounding KCs ($n = 13;14$ from left to right, $t_{25} = 3.12$, $p = 0.004$), but not outside MB ($n = 13;14$ from left to right, $t_{25} = 0.55$, $p = 0.58$). Following unpaired protocol, pyruvate levels in cortex glia within and outside MB region showed strong

correlation, which was lost following associative learning protocol (Pearson's correlation analysis; unpaired: $p = 0.0013$; $R^2 = 0.62$; associative: $p = 0.69$; $R^2 = 0.01$). **c** In flies where *Dh44-R1* was knocked down in cortex glia, no difference was observed in cortex glia pyruvate levels following associative or unpaired training protocols, neither around KCs ($n = 14;13$ from left to right, $t_{25} = 0.59$, $p = 0.55$) nor outside MB ($n = 14;13$ from left to right, $t_{25} = 0.06$, $p = 0.95$). Both after associative and unpaired protocols, pyruvate levels within and outside MB region remained significantly correlated (Pearson's correlation analysis; unpaired: $p = 0.023$; $R^2 = 0.36$; associative: $p = 0.0016$; $R^2 = 0.61$). RNAi line JF03208 (*Dh44-R1*) was used in this figure. Data are represented as mean ± SEM. ns: not significant, $p > 0.05$, **$p < 0.01$, by two-tailed Student's *t* test (**b**,**c**). On correlation plots, the red line shows the best linear regression fit to the data, and the black lines delimit the 95% confidence band. Source data are provided as a Source Data file.

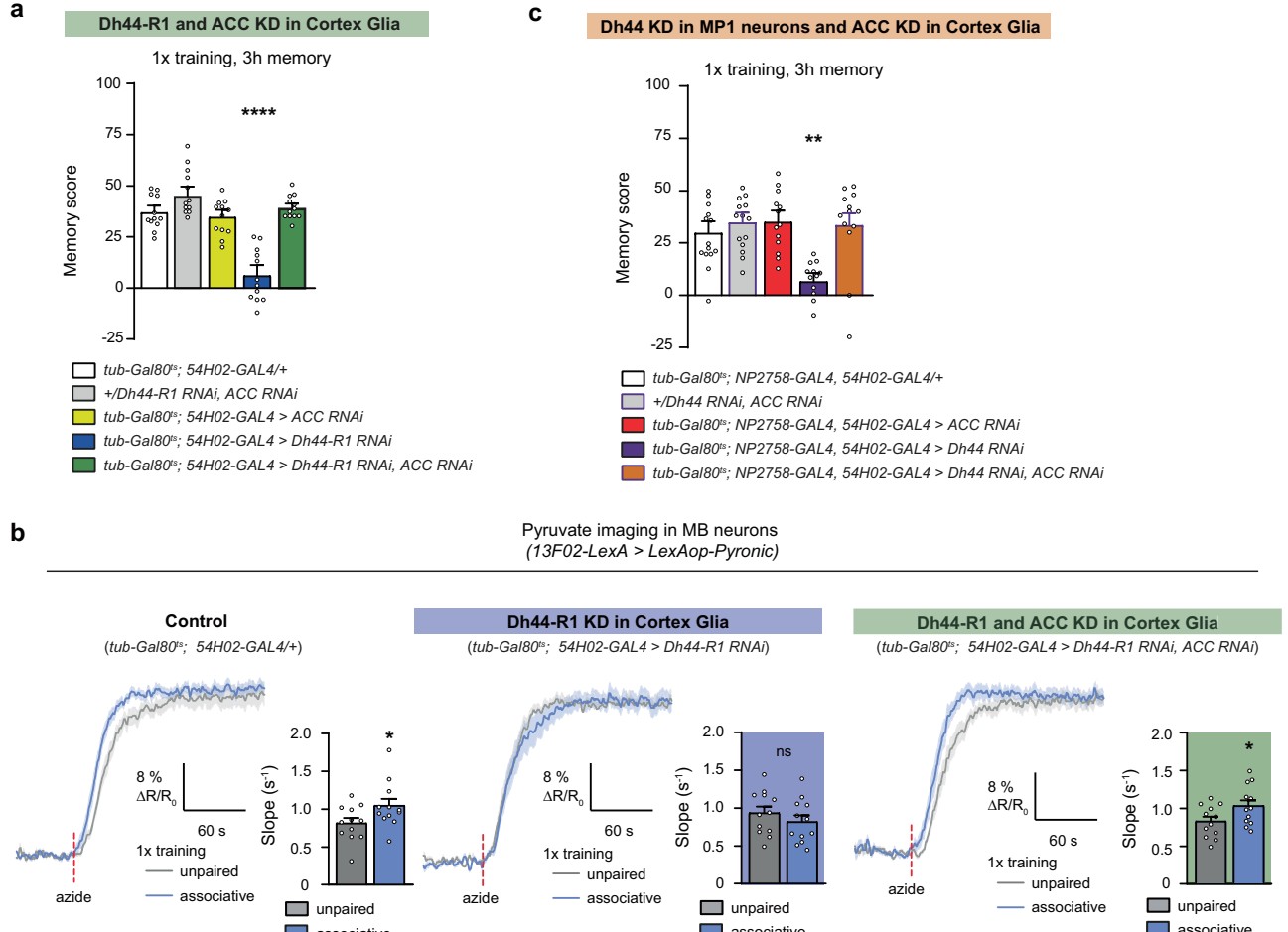

**Fig. 5 | ACC inhibition in cortex glia rescues the memory impairment induced by defective Dh44 signalling. a** The dual *Dh44-R1* and *ACC* KD in cortex glia rescued the memory deficit observed upon the single KD of *Dh44-R1* ($n = 12$, $F_{4,55} = 14.53$, $p = 4 \times 10^{-8}$). **b** Single-cycle training elicited a faster pyruvate accumulation in MB neuron axons following azide application (5 mM) as compared to non-associative unpaired conditioning ($n = 12$, $t_{22} = 2.07$, $p = 0.04$). This effect was lost in the single *Dh44-R1* knockdown condition ($n = 12$, $t_{22} = 0.96$, $p = 0.34$), but rescued upon dual *Dh44-R1* and *ACC* KD in adult cortex glia ($n = 12$, $t_{22} = 2.12$, $p = 0.04$). **c** A combination of two GAL4 drivers was used to target both MP1

neurons and cortex glia at the adult stage. As expected, *Dh44* KD in both MP1 neurons and cortex glia caused a memory defect, which was rescued in the condition of dual *Dh44* and *ACC* KD ($n = 13$, $F_{4,62} = 5.29$, $p = 0.001$). RNAi lines KK108591 (*Dh44-R1*, panel A), JF03208 (*Dh44-R1*, panel B), GD3482 (*ACC*) and JF01822 (*Dh44*) were used in this figure. Data are represented as mean ± SEM. ns: not significant, $p > 0.05$, *$p < 0.05$, **$p < 0.01$, ****$p < 0.0001$ by two-tailed Student's t test (**b**) or Tukey's pairwise comparison following one-way ANOVA (**a**, **c**). Source data are provided as a Source Data file.

memory formation in the case of an unpaired conditioning protocol[63]. Testing these candidate mechanisms experimentally would however strongly benefit from a more direct way to dynamically monitor Dh44 release. Since cortex glia is exclusively in contact with neuronal cell bodies and we observed a drop in Dh44 content in cell bodies of MP1

neurons after conditioning, it is possible that Dh44 is secreted by somatic release. However, our data neither exclude that Dh44 could be released from remote terminals, nor that long-range diffusion could allow the Dh44 neuropeptide to reach the cell body region in order to bind cortex glia receptors.

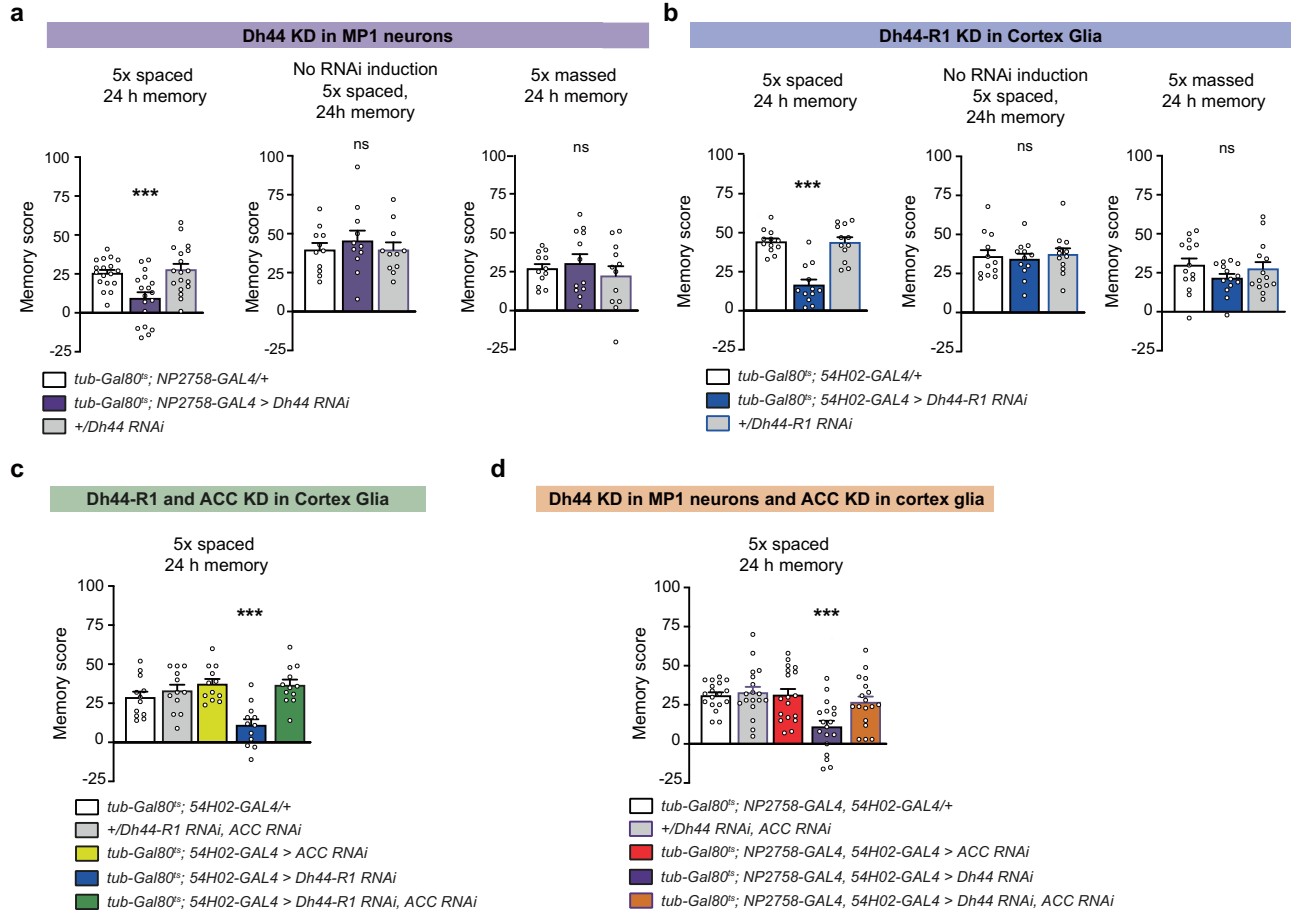

**Fig. 6 | The MP1 neuron to cortex glia Dh44 signalling axis is necessary for long-term memory. a** *Dh44* KD in MP1 neurons impaired LTM measured after spaced training ($n = 18$, $F_{2,51} = 9.22$, $p = 0.0004$) but did not affect memory resulting from massed training ($n = 12$, $F_{2,33} = 0.59$, $p = 0.55$). Flies showed normal LTM after spaced training when RNAi expression was not induced ($n = 11$, $F_{2,30} = 0.37$, $p = 0.68$). **b** *Dh44-R1* KD in cortex glia impaired LTM measured after spaced training ($n = 12$, $F_{2,33} = 1.03$, $p = 1 \times 10^{-7}$) but did not affect memory resulting from massed training ($n = 12$, $F_{2,33} = 0.66$, $p = 0.67$). Flies showed normal LTM after spaced training when RNAi expression was not induced ($n = 10$, $F_{2,27} = 1.59$, $p = 0.22$). **c** The dual *Dh44-R1* and *ACC* KD in cortex glia rescued the LTM deficit after spaced training observed upon the single KD of *Dh44-R1* ($n = 12$, $F_{4,85} = 6.94$, $p = 0.000016$). **d** *Dh44* KD in both MP1 neurons and cortex glia caused an LTM defect after spaced training, which was rescued in the dual *Dh44* and *ACC* KD condition ($n = 18$, $F_{4,85} = 6.35$, $p = 0.0002$). RNAi lines KK108591 (*Dh44-R1*), GD3482 (*ACC*) and JF01822 (*Dh44*) were used in this figure. Data are represented as mean ± SEM. ns: not significant, $p > 0.05$, ***$p < 0.001$ by Tukey's pairwise comparison following one-way ANOVA. Source data are provided as a Source Data file.

Finally, the activity of MP1 neurons is required for LTM formation, through sustained slow calcium oscillations in the hours following spaced training[33,36]. This induces a long-lasting enhancement of mitochondria metabolic activity after spaced training in the axons of α/β KCs[14,36]. As this metabolic regulation is critically dependent on dop1R2 and downstream Gq/PKCδ signalling in α/β KCs[14,36], this latter role of MP1 neurons in the metabolic tuning of α/β KCs is clearly dopamine-mediated, probably through dopamine release at the level of the peduncle. Therefore, contrasting with their local synaptic action on γ KCs, MP1 neurons also act as metabolic regulators for α/β KCs, both indirectly through neuropeptide signalling on cortex glia, and directly through dopamine/dop1R2 signalling in the peduncle area for LTM, specifically. These observations reinforce the view that the action of these versatile 'dopamine' neurons extends far beyond mere dopamine signalling. This conclusion should not be restricted to *Drosophila*, since expression and release of CRH by a fraction of mouse midbrain dopamine neurons in the ventro-tegmental area (VTA) was recently reported, which is relevant for anxiety-related behaviour[69]. As VTA dopamine neurons are involved in memory consolidation[70], our results emphasise the importance of examining how CRH from the VTA might regulate hippocampus-dependent memory processes.

## PKA activity in cortex glia downregulates FA synthesis

At the molecular scale, our results show that PKA mediates the inhibitory effect of Dh44 signalling on FA synthesis, suggesting that PKA could inhibit the key lipogenic enzyme ACC. AMPK, a master sensor of cellular energy state, is a canonical inhibitor of ACC[71], and it is well-known that PKA can indirectly, through the AMPK upstream kinase LKB1, positively regulate AMPK activity. Alternatively PKA could inhibit ACC activity by direct phosphorylation[72-74]. In a previous work in *Drosophila*, we however showed that AMPK is activated in cortex glia only in case of starvation, hence promoting LD degradation and FA oxidation[15]. In contrast, when flies were in a satiated state – as is the case in the present study – knockdown of AMPK did not increase LD content in cortex glia[15]. This suggests that AMPK is not involved in ACC inhibition by Dh44-R1 signalling, and argues in favour of an AMPK-independent mechanism through which PKA could inhibit ACC. Biochemical studies of ACC regulation identified direct phosphorylation sites by PKA in the ACC sequence in mammals (Ser[77] and Ser[1200])[72], that are partly distinct from AMPK-mediated phosphorylation (in Ser[79] and Ser[1200]). Protein sequence alignment (Supplementary Note 2) reveals that these sites are conserved in *Drosophila* ACC. Although the physiological relevance of PKA-mediated ACC phosphorylation seems tissue-dependent[74,75], it has not been studied in the brain, and our

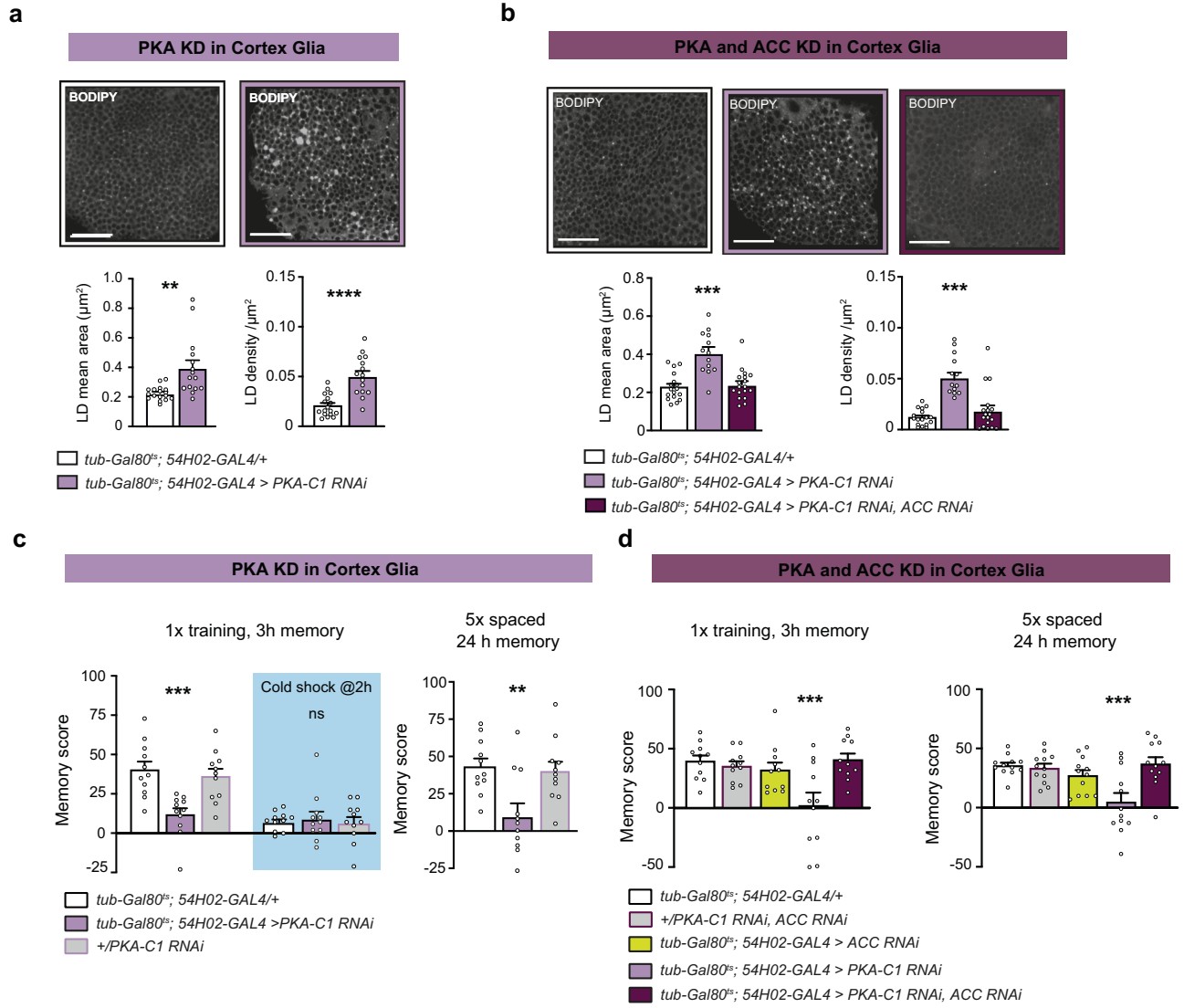

**Fig. 7 | PKA mediates Dh44-R1 inhibitory action on fatty acid synthesis in cortex glia. a** BODIPY LD staining in the MB cortex region and quantification comparing flies expressing the *PKA-C1* RNAi in adult cortex glia with a genotypic control. RNAi-expressing flies had larger LD ($n = 17;15$ from left to right, $t_{30} = 3.48$, $p = 0.001$) and higher LD densities ($n = 17;15$ from left to right, $t_{30} = 5.45$, $p = 7 \times 10^{-7}$). **b** BODIPY LD staining in the MB cortex region and quantification comparing flies expressing the *PKA* RNAi alone in cortex glia and flies expressing both *PKA-C1* RNAi and *ACC* RNAi in adult cortex glia with a genotypic control. Inhibition of *PKA-C1* expression led to an increase in LD mean area ($n = 17;13;18$ from left to right, $F_{2,45} = 18.76$ $p = 1 \times 10^{-6}$) and density ($n = 17;13;18$ from left to right, $F_{2,45} = 18.19$, $p = 2.10$) that was rescued by co-expressing *ACC* RNAi. (**c**) *PKA-C1* KD in

cortex glia impaired the total memory measured 3 h after single-cycle training ($n = 11$, $F_{2,30} = 10.44$, $p = 0.0003$), but not cold shock-resistant memory ($n = 11$, $F_{2,30} = 0.14$, $p = 0.86$). *PKA* KD in adult cortex glia also impaired LTM measured after spaced training ($n = 11$, $F_{2,30} = 6.71$, $p = 0.0039$). **d** The dual *PKA-C1* and *ACC* KD in cortex glia rescued the memory measured 3 h after single-cycle training ($n = 11$, $F_{4,50} = 5.81$, $p = 0.0006$) and LTM deficit after spaced training observed upon the single KD of *PKA* ($n = 12$, $F_{4,65} = 6.67$, $p = 0.0002$). RNAi lines JF01188 (*PKA-C1*) and GD3482 (*ACC*) were used in this figure. Data are represented as mean ± SEM. Scale bars indicate 20 μm. ns: not significant, $p > 0.05$, *$p < 0.05$, **$p < 0.01$, ****$p < 0.0001$ by two-tailed Student's *t* test (**a**, **b**) or Tukey's pairwise comparison following one-way ANOVA (c,d). Source data are provided as a Source Data file.

study suggests it might be a conserved mechanism of experience-dependent metabolic tuning of glial cells, distinct from the state-dependent AMPK-mediated regulation.

### Is CRH signalling involved in astrocyte-neuron metabolic coupling during memory in rodents?

Our findings unveil in *Drosophila* how the stress hormone response induced by associative learning drives a metabolic switch within glial cells that hold the 'metabolic key' to unlock memory formation. How could this translate to mammals? Dh44 shares 40% sequence homology with mammalian CRH[39], as well as many functional similarities[76], such that it is considered the functional homologue of CRH in invertebrates. Because of its involvement in stress response[77], the role of CRH in memory in rodents has mostly been investigated in memory

paradigms involving stressful components, i.e., fear learning. Several studies have consistently reported that short-term stress has a positive impact on memory formation and synaptic plasticity, especially in hippocampus-dependent tasks (ref. 78 and references therein). This effect might involve CRH/CRH-R1 signalling[78,79], although the underlying mechanism remains unclear.

Previously, it was shown in a passive avoidance task in rats (i.e., an aversive associative protocol inducing fear memory) that injection of either of the two main acute stress response hormones, CRH or noradrenaline (NA), had a facilitative effect on memory formation, likely through the same mechanism[80]. A direct effect of CRH on astrocytes remains to be established, but in a more recent study using the same paradigm, NA activation of β2-adrenergic receptor in hippocampal astrocytes was shown to be required for LTM formation[81].

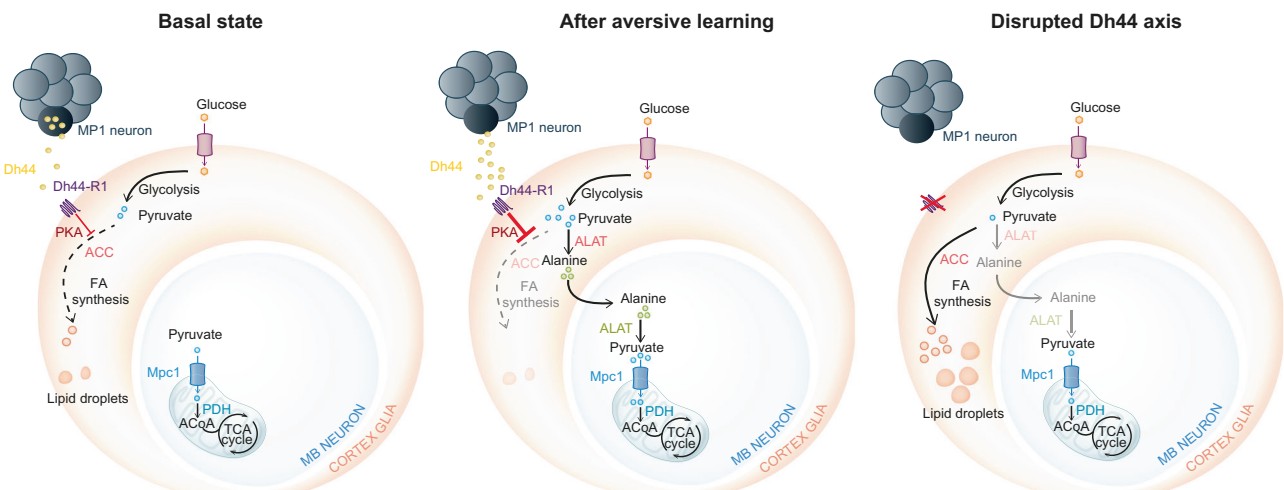

**Fig. 8 | Dh44 signalling from MP1 neurons tunes glial pyruvate routing.** Schematics illustrating the metabolic action of Dh44 signalling on cortex glia in the three conditions studied in the present work. Left: basal Dh44 release limits fatty acid synthesis in glia, through PKA activity, to levels that induce moderate LD formation, while glial support is dispensable for pyruvate supply to neuronal mitochondria. Middle: after flies learn of a danger-predictive cue, the enhanced metabolic activity of MB neuronal mitochondria is required to form threat-avoidance memory. Dh44 release by MP1 neurons is acutely enhanced. The resulting shutdown of ACC-mediated fatty acid synthesis makes glial pyruvate available for ALAT-mediated alanine conversion and export to neighbouring MB neurons. Right: when Dh44 signalling is shut down (genetic inhibition of Dh44 in MP1 neurons or Dh44-R1 in cortex glia), glial pyruvate massively flows into fatty acid synthesis, which results in extended LD production in glia and prevents metabolic support to neurons, thus preventing memory formation.

Nevertheless, a direct effect of CRH on astrocyte metabolism remains to be established, although CRH receptor expression has been reported in astrocytes in the forebrain[82], as well as in cultured rat astrocytes[83]. A starting point to extend our findings to rodents would be to assess the role of CRH-R1 in hippocampal astrocytes in memory, which to our knowledge has not been done yet.

### Fatty acid synthesis inhibition in glial cells as an energy-sparing mode of memory formation

An increasing number of essential functions fulfilled by astrocytes in mammals, or glial cells in invertebrates, are related to fatty acid metabolism and lipid homeostasis, of which de novo fatty acid synthesis is a major component. Destinations for newly synthesised fatty acids can be multiple. According to a recent study, β-oxidation of fatty acids can occur in mammalian astrocytes to bias the mitochondrial respiratory chain towards increased ROS production[18], which can have a beneficial role in cognition-relevant neuronal processes[84,85]. Fatty acids can also be embedded, together with neuron-derived peroxidated lipids[19–21], in LDs, which was shown to be a way to avoid oxidative damage to neurons. Importantly, LDs may also constitute energy stores to be used during prolonged fasting[15].

However, fatty acid biosynthesis is primarily fuelled by glucose-derived carbon atoms, although glucose is also the primary energy source for neuronal activity. In a satiated state, despite glucose being plentiful, neurons and glial cells do not consume available glucose independently. Rather, both cell types are metabolically coupled, with glial cells providing glucose-derived substrates (lactate or alanine, both issued from pyruvate) for the oxidative metabolism of neuronal mitochondria. Combined, these elements show that glial pyruvate is at stake in a competition between an anabolic need for fatty acid synthesis in glia and the catabolic needs of neighbouring neurons, which are largely driven by ongoing electrochemical activity and cognitive processes. We showed here that Dh44 signalling in cortex glia constitutively limits the fatty acid synthesis pathway. The fact that this regulatory peptidergic axis relies on learning-activated neurons allows for a dynamic, activity-dependent tuning of fatty acid synthesis inhibition, thereby making extra pyruvate available for export to neurons.

Because of the high burden they impose on the whole body energy budget, brains have evolved energy-efficient information coding schemes[56], as well as energy-saving mechanisms for when the body is under nutritional stress[86,87]. Both in adult and larval *Drosophila*, regulatory mechanisms were describe that allows adapting memory persistence to energy availability or the nutritional status[15,86,88]. Importantly, the present study was performed on fully satiated animals that were raised without any limits on their food consumption. Even under that condition, our results suggest that the formation of a transient memory lasting a few hours involves a trade-off between two competing fates for a particular metabolite, glial pyruvate. Shutdown of glial anabolism as a way to dampen variations in glucose consumption by the brain thus appears to be an additional mechanism of energy sparing by the brain. The fact that it occurs in a context of nutritional abundance highlights that energy efficiency is an integral part of the brain functional architecture, and further suggests that it is a design principle that has presided over the evolution of complex brains.

## Methods
### Experimental animals

Flies (*Drosophila melanogaster*) were raised on standard fly food (inactivated yeast 6% w/v; corn flour 6.66% w/v; agar 0.9% w/v; methyl-4hydroxybenzoate 22 mM) on a 12-h light/dark cycle at 18 °C with 60% humidity (unless mentioned otherwise). All experiments in this study were performed on 1–4-day-old adult flies. Both male and female flies were used for behaviour experiments. Female flies were used for imaging experiments because of their larger size. Canton-S strain was used as the wild type strain. All lines were outcrossed for five generations to flies carrying a CS wild type background. RNAi lines used in this study were KK108591 (RRID:Flybase_FBst0482273) and JF03208 (RRID:BDSC_28780) for *Dh44-R1*; JF03289 (RRID:BDSC_29610) for *Dh44-R2*; JF01822 (RRID:BDSC_25804) and KK110160 (RRID:Flybase_FBst0480283) for *Dh44*; GD3482 (RRID:Flybase_FBst0470958) for *ACC*; JF01188 (RRID:BDSC_31599) and JF01218 (RRID:BDSC_31277) for *PKA-C1*.

Other fly lines were MB320C (RRID:BDSC_68253), NP2758-GAL4 (RRID:DGGR_104313), tub-GAL80ts (RRID:BDSC_7019), 13F02-LexA

(RRID:BDSC_52460), 54H02-GAL4 (RRID:BDSC_45784), VT30559-GAL4 (RRID:Flybase_FBst0486483), Dh44-GAL4 (FBtp0129630), LexAop-Pyronic (FBtp0165803) and UAS-Pyronic (FBtp0141330).

## Olfactory conditioning and memory test

We used the TARGET system to restrict UAS/GAL4-mediated expression to the adult stage and not during development. Adult flies were kept at 30.5 °C for 3 days before conditioning to release GAL4 activity and achieve the induction of RNAi expression.

All behaviour experiments, including the sample sizes, were conducted similarly to previous studies from our research group[13,15,53]. Memory assays were performed on group of mixed-sex flies. Experiments reported were performed during the light phase of the cycle, but not at a specific time of the day. 20−50 flies were subjected to one of the following olfactory conditioning protocols: a single cycle (1x training), five consecutive associate training cycles (5x massed), or five associative cycles spaced by 15-min inter-trial intervals (5x spaced). For conditioning, we used custom-built barrel-type machines that allow the parallel training of up to 6 groups. Throughout the conditioning protocol, each barrel was plugged into a constant airflow at 2 L·min$^{-1}$. The sequence of one cycle of conditioning consisted of an initial 90-s period of non-odorised airflow, followed by 60 s of the conditioned odour paired with 12 pulses of 1.5 s of 60 V electric shocks. Then, after 45 s of non-odorised airflow, the second odour was presented for 60 s without electroshocks, followed by 45 s of non-odorised airflow. The odorants 3-octanol and 4-methylcyclohexanol diluted in paraffin oil to a final concentration of $2.79 \times 10^{-1}$ g.L$^{-1}$ were alternately used as the conditioned stimuli. For the unpaired conditionings used as controls in the pyruvate imaging experiments, the odour and shock stimuli were delivered separately in time, with shocks occurring 3 min before the first odorant.

Flies were kept on standard medium between conditioning and the memory test, either at 25 °C for flies tested 3 h after training or at 18 °C for flies tested 24 h after training. To test for anaesthesia-resistant memory after 1x training, flies were subjected to cold treatment exposure (4 °C for 2 min) 1 h before testing.

The memory test was performed in a T-maze apparatus, typically 3 h after single-cycle training, or 24 h after massed or spaced training. Each arm of the T-maze apparatus was connected to a bottle containing 3-octanol or 4-methylcyclohexanol, diluted in paraffin oil to a final concentration identical to the one used for conditioning. Flies were exposed simultaneously to both odorants for 1 min under darkness. The performance index was calculated as the number of flies attracted to the unconditioned odour minus the number of flies attracted to the conditioned odour, divided by the sum of the two numbers. To avoid giving disproportionate statistical importance to a small number of flies, measurements involving less than 6 flies in total were discarded. A single memory score value is the average of the performance indices from two groups of flies of the same genotype trained with either 3-octanol or 4-methylcyclohexanol as the conditioning stimulus. The indicated 'n' is the number of independent memory score values for each genotype.

## Shock avoidance

The shock response was tested for 1 min at 25 °C under darkness by placing flies in two connected barrels; electric shocks were provided in only one of the barrels. The compartment where the electric shocks were delivered was alternated between two consecutive groups. Shock avoidance scores were calculated as the number of flies choosing the non-shocked compartment minus the number of flies choosing the shocked compartment divided by the total number of flies.

## Odour avoidance

Because the delivery of electric shocks can modify olfactory acuity, we performed an olfactory avoidance test. Flies were presented an odour (3-octanol or 4-methylcyclohexanol) paired with electric shocks. Then, flies were exposed simultaneously to the T-maze, in which one arm was connected to a bottle with the non-shocked odour and the other arm was connected to a bottle with paraffin oil only. Flies were given the choice between the two arms during 1 min. The position of the odour and paraffin oil was switched in each experiment. Olfactory avoidance scores were calculated as the number of flies choosing the paraffin oil minus the number of flies choosing the non-shocked odour divided by the total number of flies.

## Alanine feeding

Flies were transferred on regular medium supplemented with L-alanine (Sigma-Aldrich, cat. no. 05129) to a final concentration of 60 mM for 72 h at 30.5 °C (to achieve RNAi induction). The behaviour experiment was then performed as described above.

## In vivo pyruvate imaging

All in vivo imaging experiments were performed on female flies, which are preferred since their larger size facilitates surgery. Crosses for imaging experiments were raised at 23 °C to avoid expression of the RNAi during development and to increase the expression level of genetically encoded sensors. To achieve the induction of RNAi expression, adult flies were kept at 30.5 °C for 3 days before conditioning.

Data were collected indiscriminately from 30 min to 1.5 h after 1x training. A single fly was placed on a plastic coverslip using a non-toxic glue and the head capsule was opened to expose the brain by gently removing the superior tracheae. Artificial haemolymph solution was added on top of the cover to bathe the head. The composition of this solution was: NaCl 130 mM, KCl 5 mM, MgCl$_2$ 2 mM, CaCl$_2$ 2 mM, D-trehalose 5 mM, sucrose 30 mM, and HEPES hemisodium salt 5 mM. At the end of surgery, any remaining solution was absorbed and a fresh 90-µl droplet of this solution was applied on top of the brain.

### Measurement of pyruvate consumption rate by 2-photon intensity imaging.

Two-photon intensity imaging was performed using a Leica TCS-SP5 upright microscope equipped with a 25x, 0.95 NA water-immersion objective. Two-photon excitation was achieved using a Mai Tai DeepSee laser tuned to 825 nm. The frame rate was 1 or 2 images per second. Images were acquired using the LAS-AF software (v2.7.3, Leica Microsystems).

Flies expressed the pyruvate sensor in MB neurons via the 13F02-LexA driver in combination with LexOp-Pyronic. RNAis were expressed in cortex glia neurons using the inducible tub-GAL80$^{ts}$; 54H02-GAL4 driver, or in the MP1 neuron using the tub-GAL80$^{ts}$; NP2758-GAL4 driver.

Measurements of pyruvate consumption were performed as follows[13,14,36]: after 1 min of baseline acquisition, 10 µl of a 50 mM sodium azide solution (prepared in the same artificial haemolymph solution) were injected into the 90-µl droplet bathing the fly's brain, bringing sodium azide to a final concentration of 5 mM.

Regions of interest (ROI) were delimited by hand around each visible MB vertical lobe, and the average intensity of the mTFP and Venus channels over each ROI was calculated over time after background subtraction. The Pyronic sensor was designed so that FRET from mTFP to Venus decreases when the pyruvate concentration increases. To obtain a signal that positively correlates with pyruvate concentration, the inverse FRET ratio was computed as mTFP intensity divided by Venus intensity. This ratio was normalised by a baseline value calculated over the 30 s preceding drug injection. The slope was calculated between 10 and 70% of the plateau, while the rise time was calculated from the time of injection to the time to reach 70% of the plateau. The indicated 'n' is the number of animals that were assayed in each condition.

**Measurement of steady-state pyruvate levels by in vivo FRET-FLIM.**
Fluorescence lifetime measurements were performed using a Leica SP8 DIVE-FALCON microscope equipped with a 25x, 1.0 NA water-immersion objective coupled to a dual output Insight infrared excitation laser (Spectra Physics). 2-photon excitation of mTFP was achieved using the tunable laser beam at 825 nm. The fluorescence lifetime of mTFP was measured by time-correlated single-photon counting in the spectral range 460–500 nm. Ds-Red was excited simultaneously by the fixed 1045 nm laser beam and its emission intensity was recorded in the spectral range 570–620 nm. All data acquisition and analysis was performed with the Leica LAS-X (v3.5.7) software. For measurements in cortex glia (Fig. 4), the DsRed channel was used to delimit cortex glia surrounding KCs; in planes outside MB (no DsRed signal), all visible cortex glia was taken as ROI. When imaging KC axons (Supplementary Fig. 4) ROI was delimited by hand as described above. Fluorescence decay curves on each specified ROI were fitted with a single-exponential decay model. Photon counts ranged between $1-2 \times 10^6$ for all analyzed ROIs.

The decay time $\tau$ of the best exponential fit typically ranged between 1.7 and 2.4 ns. Values are plotted as $\Delta\tau = \tau - \tau_{min}$, where $\tau_{min} = 1.65$ ns is the minimal mTFP lifetime from an mTFP/Venus FRET pair, as measured in ref. [89] by directly fusing the two fluorophores. The indicated 'n' is the number of animals that were assayed in each condition.

## LD staining
Neutral LDs were detected using a BODIPY 493/503 non-polar fluorescent probe. Females flies carrying the tub-Gal80[fs];54H02-GAL4 cortex glia-specific driver were crossed with males carrying the specified UAS-RNAi or with CS males. Crosses were raised at 18 °C, and 1–2-day-old adult progeny were kept at 30 °C for 3 days to induce the expression of the RNAis. Twelve hours before dissection, flies were transferred to new food bottles.

Flies were anaesthetised on ice. Brains were dissected in 1X-PBS on ice and then fixed for 30 min in 4% paraformaldehyde in 1X-PBS at room temperature. Then, brains were washed three times in 1X-PBS and incubated for 30 min with 1 μM BODIPY 493/503 under darkness. After three washes in 1X-PBS, brains were mounted using Prolong Mounting Medium. Mounting and image acquisition were performed on the same day. 1024 × 1024 images were acquired in the cortex region close to the MB calyx with a Nikon A1R confocal microscope equipped with a ×100/1.40 oil-immersion objective, using NIS-Element (v4.40) software. Confocal excitation of BODIPY was achieved using a laser tuned to 488 nm. Confocal z-stacks of LDs were imported into Fiji (ImageJ 1.52p)[90] and CellProfiler Analyst Software(v3.1.19)[91] for further analyses. A single plane in the cortex region was selected and converted into an 8-bit greyscale image. A rectangular fixed specific area (84 × 80 μm) in close proximity to the calyx was selected for analysis. Because BODIPY labels the plasma membrane in addition to LD, resulting in a bright spot-like staining over a more uniform and dimmer staining corresponding to the plasma membrane, it was necessary to threshold the image to remove the plasma membrane signal. To determine the threshold needed to keep only the lipid droplet BODIPY staining in a non-arbitrary way, we set up the following procedure. For each ROI, the pixel intensity histogram of the greyscale ROI was exported from Fiji and a Gaussian fit was performed using GraphPad Prism 9.0 software ($x < 3$ and $x > 40$ values were excluded from the fit to avoid extreme values such as black pixels). The mean and SD parameters of this Gaussian fit were extracted and used to calculate the threshold value, which was set for all images to: [(mean intensity + 4×SD)/255]. This threshold was used for further analyses using CellProfiler Software (v.3.1.9). For each ROI, after thresholding and applying a size-limit object filter (0.37–1.5 μm²) based on previous LD data published in the literature[92,93], object detection and counting was performed to identify LDs. For each ROI, the area of each identified LD

was calculated and expressed in μm² and used to calculate the mean area of LDs per ROI. For each brain, an ROI from each hemisphere was analysed and the results from both hemispheres were averaged. For a few cases in which only one hemisphere was properly visible for quantification, only one ROI was used for analysis. The indicated 'n' corresponds to the number of brains analysed.

## Quantitative PCR analyses
Quantitative PCR analyses were performed to confirm the expression of Dh44-R1 in cortex glia. Flies carrying the 54H02-GAL4 driver were crossed to the specified UAS-RNAi Dh44-R1, or with CS males. Total RNA was extracted from 50 female heads using the RNeasy Plant Mini Kit (Qiagen). Samples were reverse-transcribed with oligo(dT)20 primers using the SuperScript III First-Strand kit (Life Technologies) according to the manufacturer's instructions. The level of the target cDNA Dh44-R1 was compared against the level of α-Tub84B (Tub, CG1913) cDNA, which was used as a reference. Amplification was performed using a LightCycler 480 (Roche) and the SYBR Green I Master mix (Roche). Reactions were carried out in triplicate. The specificity and size of amplification products were assessed by melting curve analyses. Expression relative to the reference is calculated as the ratio of $2^{-\Delta Cp}$, where Cp is the crossing point. The forward primer CTGATGAGGGTGAGAACCA and the reverse primer TTTCATCGCACTCCAGCCTT were used for Dh44-R1. The forward primer ATCCAGGTGAGCAGTGTGTG and the reverse primer GTCGGGGCTACTTGTTGTGA were used for PKA-C1.

## Immunohistochemistry experiments
Female flies carrying either MB320C or the Dh44-GAL4 driver were crossed to males carrying the *UAS-mCD8::GFP* construct. Prior to dissection, 2–4-day-old female flies were fixed in 4% paraformaldehyde in PBST (PBS containing 1% Triton X-100) at 4 °C overnight. For experiments performed after training, flies were subjected to a single-cycle training as described above and fixed in 4% paraformaldehyde in PBST at different time points ($t = 0$, $t = 30$ min, $t = 60$ min, $t = 180$ min).

Fly brains were dissected on ice in PBS solution and rinsed three times for 20 min in PBST. Then, brains were blocked with 2% BSA in PBST for 2 h and incubated with primary antibodies. The following primary antibodies were used: 1:1000 rabbit anti-Dh44 (ref. 39, kindly provided by Dr. Jan Veenstra), 1:400 rat anti-elav (DSHB 7E8A10), and 1:400 mouse anti-GFP (Invitrogen, A11122). The incubation with primary antibodies was performed in the blocking solution (2% BSA in PBST) at 4 °C overnight. The following day, brains were washed 3 times for 20 min with PBST and then incubated for 3 h at room temperature with the appropriate secondary antibody diluted in blocking solution. The following secondary antibodies were used: 1:400 anti-rabbit conjugated to Alexa Fluor 594 (Invitrogen, A11037), and 1:400 anti-mouse conjugated to Alexa Fluor 488 (Invitrogen, A11029). Brains were then rinsed once in PBST for 20 min, and twice in PBS for 20 min. After rinsing, brains were mounted using Prolong Mounting Medium. Acquisitions were made with a Nikon A1R confocal microscope with a 40x/1.15 water-immersion objective or a 100x/1.40 oil-immersed objective, using NIS-Element (v4.40) software.

## Statistical analyses
All data are expressed as the mean ± SEM. For memory behaviour experiments, one experimental replicate ($n = 1$) was defined as follows: 2 groups of ~30 flies of a given genotypic condition were reciprocally conditioned using either octanol or methylcyclohexanol as the CS +. The memory score was calculated from the performance of two groups measured as described above. For imaging experiments (in vivo imaging, LD staining or immunostaining), one experimental replicate ($n = 1$) corresponds to the measurement performed on the brain of a single. Statistical analysis was performed using GraphPad Prism v9.0 software. Comparisons between two groups were

performed by unpaired Student's *t* test, and the results are given as the value $t_x$ of the t distribution, where x is a number of degrees of freedom. Comparisons among three groups were performed using a one-way ANOVA test followed by Tukey's method for pairwise comparisons between the experimental groups and their controls (significance was set at $p < 0.05$). ANOVA results are presented as the value of the Fisher distribution $F_{(x,y)}$, where x is the number of degrees of freedom between groups and y is the total number of degrees of freedom for the distribution. Asterisks in each figure refer to the least significant post hoc comparison between the genotype of interest and the genotypic controls. The nomenclature used corresponds to $*p < 0.05$; $*p < 0.01$; $***p < 0.001$, $****p < 0.0001$; ns: not significant, $P > 0.05$. Correlations were analyzed by computing the Pearson's correlation analysis.

### Reporting summary

Further information on research design is available in the Nature Portfolio Reporting Summary linked to this article.

## Data availability

No datasets that require mandatory deposition into a public database were generated during the current study. Processed data from imaging experiments and raw data of behavioural assays, are provided in the Source data file. Unprocessed images, which represent a large volume, were not deposited in a public repository, as permanent storage by data centres raises increasing environmental and energy concerns. These data are available through e-mailing the corresponding authors, and will be shared without restriction within a week. Source data are provided with this paper.

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

## Acknowledgements

We thank Dr Jan Veenstra for generously sharing the Dh44 antibody, and Dr Monica Dus for the Dh44-GAL4 line, as well as Christelle Beauchamp and Alexandre Didelet for fly food preparation. We are also grateful to Dr Dafni Hadjieconomou and Dr Jaime de Juan Sanz for their critical comments on the manuscript. This work was funded by grants from the ERC (ERC-AdG-741550, to T.P.), ANR (ANR-20-CE92-0047-01, to P.-Y.P.), the Fédération pour la Recherche sur le Cerveau (FRC, to P.-Y.P), Labex Memolife (to P.-Y.P.), and by a DIM ELICIT's equipment grant from Région Ile-de-France (to P.-Y.P).

## Author contributions

R.F. and Y.R. collected and analyzed data. P.-Y.P. and T.P. designed and supervised the project, and collected funding. P.-Y.P. wrote the manuscript, R.F. and Y.R. prepared the figures. All autors edited the manuscript.

## Competing interests

The authors declare no competing interests.
