## [Transparent Peer Review file · Nature Communications]

Diverting glial glycolytic flux towards neurons is a memory-relevant role of *Drosophila* CRH-like signalling

Corresponding Author: Dr Pierre-Yves Plaçais

Version 0:

Reviewer comments:

Reviewer #1

(Remarks to the Author)

This is a very well performed and nicely written study in which Frances and colleagues report on the link between *Drosophila* neuropeptide Dh44, which is the mammalian CRH homolog, with its inhibitory role on glial lipid synthesis, thus indirectly leaving spared pyruvate (via alanine) neuronal fueling. Furthermore, this effect is amenable to upregulation by learning, hence connecting glial-neuronal metabolic cooperation with memory formation. At the light of their results, the authors conclude that this mechanism show a competition between glial anabolism and neuronal energy fueling. The experimental design, strategy, methodology and statistical analysis are well executed, giving rise to an elegant piece of work.

Comments

1. Whilst the interaction between metabolic pathways and behavioral responses are studied using validated genetic strategies, under the point of view of this reviewer there is a missing molecular link between the Dh44-R1 receptor activation and ACC constitutive inhibition. Given the critical relevance of this interaction in the whole metabolic fueling sparing model herein described, it would be helpful if the authors could provide any experimental piece of evidence to point out such a molecular link.
2. Line 402 – Besides ref. 77, please cite also PMID 32694785 given that this work more specifically deals with the message that astrocytic mitochondrial ROS plays a role in cognition.
3. The discussion section may be too large -6 pages, i.e. roughly the same extension than the Results section text. Even if there is no words limitation, on several occasions some concepts are over-interpreted leading to excessive speculation. For instance, the sub-section ranging from line 357 to 395 largely exceeds the usually acceptable speculative levels. Whilst it is usually welcome to show the potential implications of these results into mammals, under the point of view of this reviewer, this subsection goes too far away in this task. The authors should significantly simplify it whilst obviously keeping the same extension to mammals. In other sub-sections, whilst the grammar is correct and logically structured, some simplification and shortening of the messages might help for the reader to follow up the story. This is particularly relevant having into account the very large introductory paragraph (2.5 pages) that already contains some of the information discussed in the Discussion section.

Reviewer #2

(Remarks to the Author)

In recent years, the laboratory of Preat and Plaçais has intensively investigated the connection between cellular energy consumption, metabolic signaling pathways and the formation of longer-term memory phases. They use the exquisitely suitable model organism *Drosophila melanogaster*, as the cellular connections that mediate associative memory formation are well known. This research has led to a whole series of highly interesting, high-quality publications. The present

manuscript is part of this series and is in the same context. The study by Frances et al reports that a specific type of glia cell (cortex glia) has an influence on neurons involved in memory formation through energy supply. Furthermore, this process is modulated by a peptide hormone (Dh44) secreted by a small group of dopaminergic neurons that typically signals the unconditioned stimulus. The last aspect is quite novel, highly interesting and absolutely worth reporting.

The manuscript is very well written, easy to follow and logical in its arguments. The experiments are well conceived, the methods are state of the art and all necessary controls are at place. The statistics are sound, the figures illustrate the findings overall in a convincing manner, and the discussion section is an interesting and stimulating read. I recommend publication of the report, but would like to add some points the authors might want to take into consideration.

1. Line 121: FA is not defined (fatty acid?).

2. The authors use the term "inhibition" throughout the text to describe RNAi-mediated downregulation of genes. This is sometimes highly confusing, because the term inhibition can imply different things (hyperpolarization, block of transmitter release, receptor antagonist, etc.). Short descriptions like "Inhibition of Dh44-R2" (line 871), as an example, or "ACC being inhibited" (line 133) is insufficient and causes such confusion. I recommend to use RNAi-mediated knockdown or RNAi-mediated downregulation throughout to make clear what the experimental manipulation actually was.

3. The anatomical illustrations in Figure 1 do not clearly indicate where exactly the cells shown are located in the *Drosophila* brain, e.g. relative to the mushroom body. It would be very helpful if, for example, an overview image of the brain (not only a sketch as in extended figure 1) was included in which the section used for the BODIPY LD stainings was marked. The length of the scaling bars should also be indicated. The bar diagrams indicate means and SEM, I assume? This indication occurs in extended figures, but not in the main figures.

4. I did not fully understand how the authors envisage that the depletion of Dh44 in MP1 neurons occurs when an odor is associated with a shock, but not if both stimuli are presented in separation (figure 4C). Doesn't this implicate a coincidence detection mechanism in MP1 neurons? Or can this rely on synaptic feedback from Kenyon cells (i.e., through reciprocal synapses? After all, dopaminergic neurons innervating the mushroom body heel and gamma 1-3 compartments (as MP1) respond also to odors (Riemensperger et al., *Curr Biol.* 2005 Nov 8;15(21):1953-60).

5. Do the authors have any idea how their proposed role of cortex glia cells relates to the recent report by Miyashita et al. (*Science.* 2023 Dec 22;382(6677):eadf7429) on the role of glia in aversive conditioning?

6. The authors elaborate nicely on the dependence of different memory phases (ARM, LTM) on energy supply. This finding has been confirmed in larval *Drosophila* as well, and the respective publication (Eschment et al., *PLoS Genet.* 2020 Oct 26;16(10):e1009064) I think deserves not to be ignored in the discussion.

Otherwise the manuscript is great and I congratulate the authors to their findings.

Reviewer #3

(Remarks to the Author)

Overview

This manuscript seeks to show how the brain coordinates rapid and local responses to changing energy requirements consequent of memory-related neuronal activity. Specifically, it examines neuron-glia interactions that prime glia to supply energy to neurons in the context of medium and long-term memory in *Drosophila melanogaster*. It persuasively demonstrates that the neuropeptide DH44, released from a pair of dopaminergic neurons well known in the context of multiple forms of memory, instructs cortex glia to shift away from anabolic metabolism, and instead promote the transport of alanine / pyruvate to alpha/beta Kenyon cells. Overall, this manuscript advances our understanding of the mechanisms by which memory circuits control glial metabolic programs to supply energy for memory-critical functions. However, additional evidence is needed for the claim that downregulating fatty acid synthesis in glia increases the pool of metabolites ready for export to neurons.

Novelty

Previous work from this lab has identified metabolites passed from glia to neurons in the context of various forms of memory, including alanine (Rabah et al 2023), ketones (Silva et al 2022), and glucose (de Tredern et al 2021). This work adds the identification of a neuropeptide (Dh44) released from MP1 neurons onto cortex glia that primes glial metabolism for energy export to neurons. Thus, this work is taking the broader story of glial support for neuronal memory energetic/metabolic requirements upstream. Notably, the neurons that cue the glia (MP1) are not the neuron that receives the energetic supply (alpha/ beta KCs); indeed, MP1 neurons seem to receive energetic cues and influence metabolic decisions not only in cortex glia via DH44 but in alpha/ beta KCs via dopamine (Musso, Tchenio, and Preat, 2015, Placais et al 2017). Taken together with other work in the field and from this lab, this manuscript adds neuron-glia neuropeptide signaling to the already-complex picture of how memory circuits dynamically and coordinately adjust glial and neuronal metabolism to support memory.

Summary of noteworthy results

Broad

1. A neuronal peptide regulates glial metabolism and suppresses fatty acid synthesis/ storage. Memory training stimulates release of this peptide.

2. Glial metabolism must be regulated by memory-active neurons for the correct provision of energy to a separate population of memory-consolidating neurons.

3. Within glia, there is competition of anabolic synthesis of fatty acids vs provision of catabolic supply to neurons (see Conceptual Critique 1)

Specific (*Drosophila* memory circuits)

1. This work identifies another role for MP1 in memory, and another role for MP1 in regulating metabolic decisions of other cells—this time cortex glia.
2. MP1 releases Dh44, which was previously thought to be only present in a small cluster elsewhere in the brain (the PI).
3. Neuronal Dh44 release tonically suppresses lipid droplet synthesis in cortex glia.
4. Neuronal pyruvate supply for MTM and LTM depends upon suppression of glial anabolic processes by Dh44 signaling (see critique).

Conceptual Critiques

In general, we find the claims in this manuscript are well supported by the data presented. Appropriate controls have been executed using established methods previously published from this lab. The manuscript has a good logical flow and experiments are clearly justified and interpreted. The figures are clear and easy to follow. However, the following points should be addressed to strengthen the manuscript:

1. Link between glial fatty acid synthesis, glial pyruvate levels, and alanine export

In the introduction, the authors write, "In addition to its basal activity, this neuropeptide signalling axis was transiently stimulated by aversive olfactory learning, thereby decreasing the anabolic consumption of glycolysis-derived pyruvate". The same idea is echoed in the discussion, e.g. line 432. The claim in bold, that cortex glial pyruvate rises when fatty acid synthesis is inhibited (either by ACC knockdown or by Dh44 signaling) has not been demonstrated. Demonstrating this is central to the claim that there is competition between anabolic synthesis and energy export in glia.

Relatedly, Figure 6 and the Discussion of this manuscript seek to link the downregulation of fatty acid synthesis in cortex glia with the upregulation of alanine production for transport to neurons. This work has demonstrated the link between blocking fatty acid synthesis in cortex glia and an increase in pyruvate in neurons, and another paper from this lab has shown that increases in pyruvate in neurons are downstream of alanine transport into neurons from glia. However, the link between blocking fatty acid synthesis in cortex glia via Dh44 and increasing alanine transport from cortex glia to neurons has not been persuasively demonstrated in this manuscript. We suggest at least one of the following experiments to make this link more solid, and to provide further support for the "competition" hypothesis described above. In order of preference, they are:

- Determine whether pyruvate (a detectable intermediate of the alanine synthesis pathway) increases in cortex glia upon memory training, and DH44-R1 RNAi in cortex glia and/ or DH44 RNAi in MP1 neurons prevents this increase.
 - o This experiment could also reinforce the acute requirement for Dh44 regulation of fatty acid in memory – currently, because knockdown experiments are inherently not temporally precise, there is some ambiguity between the requirement for less basal glial fatty acid synthesis in the days / hours preceding memory and the requirement for less fatty acid synthesis during memory itself. Demonstrating acute changes in glial pyruvate immediately after memory training would strongly support the acute interpretation, which seems to be favored by the authors.
- Whether alanine feeding restores memory in the context of DH44-R1 RNAi in cortex glia and/ or DH44 RNAi in MP1 neurons (a la figure 3D of Rabah et al 2023)
- (less preferable, because lacking link to memory) Basal pyruvate levels in cortex glia decline with DH44-R1 RNAi in cortex glia and/ or DH44 RNAi in MP1 neurons

2. Support for double RNAi experiments

The authors should provide greater context for a series of experiments in which two RNAis, ACC and DH44, were expressed using drivers for two separate populations, cortex glia and MP1 neurons. We totally understand the technical impossibility of doing the ideal experiment (ACC RNAi only in cortex glia and DH44 RNAi only in MP1 neurons) and generally find this less-than-ideal method an acceptable compromise. However, to support and contextualize this series of experiments, answers to the following questions are important. Publicly available sequencing data should be able to clarify many of these.

- Do neurons express ACC, since they do not form lipid droplets? Do they do so at levels comparable to glia?
- Likewise, do cortex glia express Dh44?
- If neuronal expression of ACC is low or absent, and/or cortex glia expression of Dh44 is low or absent, presenting this data (in a supplemental figure or table) could further strengthen confidence in Figure 1E, 4B, 4C, and 5D experiments that took the double RNAi (Dh44 and ACC simultaneous knockdown in cortex glia and MP1) approach. If this cannot be shown, we would suggest changing the figure headers from "Dh44 RNAi in MP1 and ACC RNAi in Cortex Glia" to "Dh44 and ACC RNAi in MP1 and Cortex Glia." Everywhere else the text is adequately clear.

3. Conceptual framing

We are not necessarily convinced of the universal applicability of the idea that overall metabolic supply to the brain is constant, which appears both in the introduction (line 46) and the discussion (line 438). However, we also do not find this claim critical for the conclusions of this paper, and think it can be removed without much loss of impact. Previous work from this lab and others has shown flies eat more (Placais et al 2017) after certain types of memory training. Would this not suggest the brain is actively driving behaviors that increase the total energy available for memory consolidation, at least on the scale of hours? Even in the presence of increased energy supply, neurons could still receive that energy supply via glia, and glia could still be required to limit the amount they divert for fatty acid synthesis. Thus, there is no need to make this rather sweeping claim about constant total energy availability to the brain, and we find that other work undermines it in the case of *Drosophila* memory.

Minor points

- The DH44 antibody was not properly validated in the Cabrero 2002 paper as far as we can see (as claimed on line 148), but we believe this manuscript has sufficiently validated it with the RNAi experiment in Figure 1C. If it were possible to mis-express Dh44 in a population that does not usually express it to serve as a positive control, that would be an encouraging further validation, but not absolutely required for the interpretation of experiments presented here.
- Methods - Please provide the exact recipe for the fly food used.

- Methods - Please clarify memory experiment methods: was there any set threshold for the number of flies making a choice for the data point to be counted? For example, if one fly chose one odor, one fly chose the other odor, and 28 flies remained in the center, would this data point be discarded, or would it be included as a PI of 0?
- Methods – Please specify the time of day memory experiments were performed (See: <https://www.biorxiv.org/content/10.1101/2024.01.25.577231v1>)
- Line 56: “fruit flies” (two words)
- Line 88: “memorize” should probably just be “remember” (“memorize” has connotations of deliberately studying something)
- Line 125: “confirmed” should be “resulted in” or “caused” or “produced” or something like that
- Line 141: it would be clearer to write “PI neurons” instead of “Dh44 neurons” since it will be soon revealed that there are more Dh44 neurons than previously thought

Reviewer #4

(Remarks to the Author)

I co-reviewed this manuscript with one of the reviewers who provided the listed reports. This is part of the Nature Communications initiative to facilitate training in peer review and to provide appropriate recognition for Early Career Researchers who co-review manuscripts

Version 1:

Reviewer comments:

Reviewer #1

(Remarks to the Author)

The authors have successfully addressed the concerns of this reviewer and, accordingly, the manuscript is now much improved.

Reviewer #2

(Remarks to the Author)

The authors have done a great job in addressing all comments and points of concern. I recommend publication of the revised manuscript.

Reviewer #3

(Remarks to the Author)

The authors have satisfactorily addressed our concerns.

Reviewer #4

(Remarks to the Author)

We would like to thank all reviewers for the overall positive evaluation of our initial manuscript, as well as their insightful comments and critiques. We believe that, by addressing the major concerns with new experiments, the revised manuscript has been much strengthened and now makes a convincing case about the existence of a competition between neuronal fueling and fatty acid synthesis in cortex glia, and the dynamic regulation of this competition by Dh44 signaling. Please see below our point-by-point response to all concerns that were raised.

Reviewer #1 (Remarks to the Author):

This is a very well performed and nicely written study in which Frances and colleagues report on the link between *Drosophila* neuropeptide Dh44, which is the mammalian CRH homolog, with its inhibitory role on glial lipid synthesis, thus indirectly leaving spared pyruvate (via alanine) neuronal fueling. Furthermore, this effect is amenable to upregulation by learning, hence connecting glial-neuronal metabolic cooperation with memory formation. At the light of their results, the authors conclude that this mechanism show a competition between glial anabolism and neuronal energy fueling. The experimental design, strategy, methodology and statistical analysis are well executed, giving rise to an elegant piece of work

Comments

1. Whilst the interaction between metabolic pathways and behavioral responses are studied using validated genetic strategies, under the point of view of this reviewer there is a missing molecular link between the Dh44-R1 receptor activation and ACC constitutive inhibition. Given the critical relevance of this interaction in the whole metabolic fueling sparing model herein described, it would be helpful if the authors could provide any experimental piece of evidence to point out such a molecular link.

Following this comment, we started to explore the molecular pathways acting downstream Dh44-R1. As this receptor is known to be a cAMP-coupled GPCR (Cabrero et al., 2002 ; Johnson et al., 2005), we tested whether PKA activity was mediating ACC inhibition. In the revised version, we include new experimental data showing that :

- (i) Knockdown of PKA-C1 (the gene encoding for the catalytic subunit of PKA) in cortex glia increased their lipid droplet content, which depends on ACC.
- (ii) Knockdown of PKA-C1 in cortex glia impaired middle-term and long-term memory
- (iii) These memory defects were rescued by simultaneous knock-down of ACC in cortex glia.

All these phenotypes recapitulate the results obtained with Dh44-R1 knockdown, strongly supporting that PKA mediates the Dh44-R1 inhibitory action on ACC. These results are shown on a new Figure 7.

AMPK is a well-established inhibitor of ACC. However, previous published data from our laboratory (Silva et al., 2022) allow to exclude a role of AMPK on lipid droplet synthesis in satiated flies. In the discussion section, we included a new paragraph elaborating on the putative mechanisms of ACC inhibition by PKA by direct phosphorylation, independently of AMPK.

2. Line 402 – Besides ref. 77, please cite also PMID 32694785 given that this work more specifically deals with the message that astrocytic mitochondrial ROS plays a role in cognition.

We thank Reviewer #1 for pointing out this error. This paper was actually the one that we intended to cite, rather than the Jimenez-Blasco et al, which is less relevant here. We now cite both articles.

3. The discussion section may be too large -6 pages, i.e. roughly the same extension than the Results section text. Even if there is no words limitation, on several occasions some concepts are over-interpreted leading to excessive speculation. For instance, the sub-section ranging from line 357 to 395 largely exceeds the usually acceptable speculative levels. Whilst it is usually welcome to show the potential implications of these results into mammals, under the point of view of this reviewer, this subsection goes too far away in this task. The authors should significantly simplify it whilst obviously keeping the same extension to mammals. In other sub-sections, whilst the grammar is correct and logically structured, some simplification and shortening of the messages might help for the reader to follow up the story. This is particularly relevant having into account the very large introductory paragraph (2.5 pages) that already contains some of the information discussed in the Discussion section.

We acknowledge that we might have pushed a bit further than usual the parallelism with mammalian brain, because there are indeed intriguing reports that would be consistent with the existence of similar mechanisms to what we discovered here in the fly brain. Reviewer #2 mentioned that 'discussion section is stimulating and interesting', hence we wish to keep the various messages that we conveyed in the initial version. Still, we strongly shortened the particular subsection that Reviewer #1 pointed out, by cutting its most speculative parts. We also removed part of discussion regarding the versatility in signaling from MP1 neurons, which probably was going into too many details not directly connected to the present study.

Please note, however, that because of the new findings about the role of PKA, we also introduced a new (short) paragraph in the discussion about the possible mechanism of PKA-mediated inhibition of ACC (lines 398-414), and a couple sentences about how Dh44 release by MP1 neurons could be activated specifically upon paired odour/shock (following Reviewer #2's comment ; lines 365-380).

Reviewer #2 (Remarks to the Author)

In recent years, the laboratory of Preat and Placais has intensively investigated the connection between cellular energy consumption, metabolic signaling pathways and the formation of longer-term memory phases. They use the exquisitely suitable model organism *Drosophila melanogaster*, as the cellular connections that mediate associative memory formation are well known. This research has led to a whole series of highly interesting, high-quality publications. The present manuscript is part of this series and is in the same context. The study by Frances et al reports that a specific type of glia cell (cortex glia) has an influence on neurons involved in memory formation through energy supply. Furthermore, this process is modulated by a peptide hormone (Dh44) secreted by a small group of dopaminergic neurons that typically signals the unconditioned stimulus. The last aspect is quite novel, highly interesting and absolutely worth reporting.

The manuscript is very well written, easy to follow and logical in its arguments. The experiments are well conceived, the methods are state of the art and all necessary controls are at place. The statistics are sound, the figures illustrate the findings overall in a convincing

manner, and the discussion section is an interesting and stimulating read. I recommend publication of the report, but would like to add some points the authors might want to take into consideration.

1. Line 121: FA is not defined (fatty acid?).

Thanks for pointing this out. We have removed 'FA' and written 'fatty acid' explicitly, as everywhere else in the manuscript.

2. The authors use the term "inhibition" throughout the text to describe RNAi-mediated downregulation of genes. This is sometimes highly confusing, because the term inhibition can imply different things (hyperpolarization, block of transmitter release, receptor antagonists, etc.). Short descriptions like "Inhibition of Dh44-R2" (line 871), as an example, or "ACC being inhibited" (line 133) is insufficient and causes such confusion. I recommend to use RNAi-mediated knockdown or RNAi-mediated downregulation throughout to make clear what the experimental manipulation actually was.

We followed this recommendation and changed all the occurrences of « inhibition » to « knockdown » or « RNAi-mediated knockdown » where relevant.

3. The anatomical illustrations in Figure 1 do not clearly indicate where exactly the cells shown are located in the *Drosophila* brain, e.g. relative to the mushroom body. It would be very helpful if, for example, an overview image of the brain (not only a sketch as in extended figure 1) was included in which the section used for the BODIPY LD stainings was marked. The length of the scaling bars should also be indicated. The bar diagrams indicate means and SEM, I assume? This indication occurs in extended figures, but not in the main figures.

As suggested, we now provide an overview image of the whole brain on Supplementary Figure 1, to better illustrate the region where the zoomed images presented on Figure 1 come from.

Regarding the SEM, we confirm that each legend in the initial version already stated « Data are represented as mean \pm SEM » at the end of the legend, together with the statistical test description.

4. I did not fully understand how the authors envisage that the depletion of Dh44 in MP1 neurons occurs when an odor is associated with a shock, but not if both stimuli are presented in separation (figure 4C). Doesn't this implicate a coincidence detection mechanism in MP1 neurons? Or can this rely on synaptic feedback from Kenyon cells (i.e., through reciprocal synapses? After all, dopaminergic neurons innervating the mushroom body heel and gamma 1-3 compartments (as MP1) respond also to odors (Riemensperger et al., *Curr Biol.* 2005 Nov 8;15(21):1953-60).

Reviewer 2 raised here a very interesting point that we omitted to discuss. Indeed, our data show that MP1 neurons respond differently to the odor/shock association than to the separated stimuli. We see three possible mechanisms that could relay the coincidence information to these neurons.

First, there could be an intrinsic coincidence detection mechanism in MP1 neurons as pointed out by Reviewer 2, given that these neurons reportedly respond both to odours and shocks.

Second, an outcome of the fly brain connectome reconstruction is that MP1 neurons are strongly postsynaptic to both gamma and alpha/beta KCs, which are known to be synergistically activated by the paired odor/shock stimulation.

Finally, a recent report (Miyashita et al., Science, 2023), reported that another type of glia cells -ensheathing glia- gates the release of dopamine to Kenyon cells by dopamine neurons (including MP1 neurons). This constitutes an appealing mechanism through which MP1 neurons would react differentially to the associative and unpaired protocols. In future studies, it will be exciting to see if Dh44 release by MP1 is dependent on ensheathing glia signaling to MP1 neurons.

In the revised version of the manuscript, we included a new paragraph addressing this highly relevant point (lines 365-380).

5. Do the authors have any idea how their proposed role of cortex glia cells relates to the recent report by Miyashita et al. (Science. 2023 Dec 22;382(6677):eadf7429) on the role of glia in aversive conditioning?

Cf previous point : although Miyashita report a role of a different type of glial cells in aversive conditioning (ensheathing glia rather than cortex glia), the mechanism they evidenced could underlie the differential release of Dh44 peptide upon associative and unpaired conditioning.

6. The authors elaborate nicely on the dependence of different memory phases (ARM, LTM) on energy supply. This finding has been confirmed in larval *Drosophila* as well, and the respective publication (Eschment et al., PLoS Genet. 2020 Oct 26;16(10):e1009064) I think deserves not to be ignored in the discussion.

We added a sentence (line 458) describing that in all *Drosophila* stages, memory formation is subjected to metabolic control, citing the work by Eschment et al.

Otherwise the manuscript is great and I congratulate the authors to their findings.

Reviewer #3 (Remarks to the Author):

Overview

This manuscript seeks to show how the brain coordinates rapid and local responses to changing energy requirements consequent of memory-related neuronal activity. Specifically, it examines neuron-glia interactions that prime glia to supply energy to neurons in the context of medium and long-term memory in *Drosophila melanogaster*. It persuasively demonstrates that the neuropeptide DH44, released from a pair of dopaminergic neurons well known in the context of multiple forms of memory, instructs cortex glia to shift away from anabolic metabolism, and instead promote the transport of alanine / pyruvate to alpha/beta Kenyon cells. Overall, this manuscript advances our understanding of the mechanisms by which memory circuits control glial metabolic programs to supply energy for memory-critical functions. However, additional evidence is needed for the claim that downregulating fatty acid synthesis in glia increases the pool of metabolites ready for export to neurons.

Novelty

Previous work from this lab has identified metabolites passed from glia to neurons in the context of various forms of memory, including alanine (Rabah et al 2023), ketones (Silva et al 2022), and glucose (de Tredern et al 2021). This work adds the identification of a neuropeptide (Dh44) released from MP1 neurons onto cortex glia that primes glial metabolism for energy export to neurons. Thus, this work is taking the broader story of glial support for neuronal memory energetic/metabolic requirements upstream. Notably, the neurons that cue the glia (MP1) are not the neuron that receives the energetic supply (alpha/ beta KCs); indeed, MP1 neurons seem to receive energetic cues and influence metabolic decisions not only in cortex glia via DH44 but in alpha/ beta KCs via dopamine (Musso, Tchenio, and Preat, 2015, Placais et al 2017). Taken together with other work in the field and from this lab, this manuscript adds neuron-glia neuropeptide signaling to the already-complex picture of how memory circuits dynamically and coordinately adjust glial and neuronal metabolism to support memory.

Summary of noteworthy results

Broad

1. A neuronal peptide regulates glial metabolism and suppresses fatty acid synthesis/ storage. Memory training stimulates release of this peptide.
2. Glial metabolism must be regulated by memory-active neurons for the correct provision of energy to a separate population of memory-consolidating neurons.
3. Within glia, there is competition of anabolic synthesis of fatty acids vs provision of catabolic supply to neurons (see Conceptual Critique 1)

Specific (Drosophila memory circuits)

1. This work identifies another role for MP1 in memory, and another role for MP1 in regulating metabolic decisions of other cells—this time cortex glia.
2. MP1 releases Dh44, which was previously thought to be only present in a small cluster elsewhere in the brain (the PI).
3. Neuronal Dh44 release tonically suppresses lipid droplet synthesis in cortex glia.
4. Neuronal pyruvate supply for MTM and LTM depends upon suppression of glial anabolic processes by Dh44 signaling (see critique).

Conceptual Critiques

In general, we find the claims in this manuscript are well supported by the data presented. Appropriate controls have been executed using established methods previously published from this lab. The manuscript has a good logical flow and experiments are clearly justified and interpreted. The figures are clear and easy to follow. However, the following points should be addressed to strengthen the manuscript:

1. Link between glial fatty acid synthesis, glial pyruvate levels, and alanine export

In the introduction, the authors write, “In addition to its basal activity, this neuropeptide signalling axis was transiently stimulated by aversive olfactory learning, thereby decreasing the anabolic consumption of glycolysis-derived pyruvate”. The same idea is echoed in the discussion, e.g. line 432. The claim in bold, that cortex glial pyruvate rises when fatty acid synthesis is inhibited (either by ACC knockdown or by Dh44 signaling) has not been demonstrated. Demonstrating this is central to the claim that there is competition between anabolic synthesis and energy export in glia.

Relatedly, Figure 6 and the Discussion of this manuscript seek to link the downregulation of fatty acid synthesis in cortex glia with the upregulation of alanine production for transport to neurons. This work has demonstrated the link between blocking fatty acid synthesis in cortex glia and an increase in pyruvate in neurons, and another paper from this lab has shown that increases in pyruvate in neurons are downstream of alanine transport into neurons from glia. However, the link between blocking fatty acid synthesis in cortex glia via Dh44 and increasing alanine transport from cortex glia to neurons has not been persuasively demonstrated in this manuscript. We suggest at least one of the following experiments to make this link more solid, and to provide further support for the "competition" hypothesis described above. In order of preference, they are:

- Determine whether pyruvate (a detectable intermediate of the alanine synthesis pathway) increases in cortex glia upon memory training, and DH44-R1 RNAi in cortex glia and/ or DH44 RNAi in MP1 neurons prevents this increase.

- o This experiment could also reinforce the acute requirement for Dh44 regulation of fatty acid in memory – currently, because knockdown experiments are inherently not temporally precise, there is some ambiguity between the requirement for less basal glial fatty acid synthesis in the days / hours preceding memory and the requirement for less fatty acid synthesis during memory itself. Demonstrating acute changes in glial pyruvate immediately after memory training would strongly support the acute interpretation, which seems to be favored by the authors.

We thank Reviewer #3 for this insightful comment, and we agree that in the initial version, the acute requirement of Dh44 signalling after training was only supported by the Dh44 staining experiments in MP1 neurons. The 'flux-stop' protocol that we have used so far in neuronal cells was intended to measure pyruvate consumption rate. Measuring steady-state pyruvate level requires measuring the absolute FRET of the sensor. To achieve this rigorously, we turned to 2-photon fluorescence lifetime imaging (FLIM) to measure FRET as reported by variation in the lifetime of the donor fluorophore of the FRET pair (FRET-FLIM). The Pyronic sensor has already been used once in FRET-FLIM mode (Diaz-Garcia et al., eLife (2021)). To further validate this approach, and since sodium azide mitochondrial blockade robustly increases pyruvate levels in neurons, we verified that this indeed resulted in an increase in mTFP lifetime (Supplementary Figure 4A).

This approach allowed us to reveal that in cortex glia, associative learning induces an increase in pyruvate levels. Moreover, this effect was observed around KCs, but not in cortex glia outside the MB region (Figure 4). Although cortex glia pyruvate levels within and outside MB area were well correlated in control flies, this correlation was completely lost after associative learning. This indicates that cortex glia around MB is subjected to a specific and independent regulation during memory formation, so that a local increase in glial pyruvate makes it available for transamination into alanine and export to neurons. As predicted by our model, Dh44-R1 knockdown in cortex glia prevented this increase to occur.

Altogether, these new data represent a major and highly relevant addition to our initial manuscript, and support the view that Dh44 signalling mediates an MB-specific metabolic regulation of cortex glia pyruvate availability following learning to fuel memory formation in MB neurons.

- Whether alanine feeding restores memory in the context of DH44-R1 RNAi in cortex glia and/ or DH44 RNAi in MP1 neurons (a la figure 3D of Rabah et al 2023)

This is an excellent suggestion to test our hypothesis. We performed these experiments, and observed that alanine feeding could indeed rescue middle-term memory in both contexts. These new data are now displayed on Figure 3 E,F.

- (less preferable, because lacking link to memory) Basal pyruvate levels in cortex glia decline with DH44-R1 RNAi in cortex glia and/ or DH44 RNAi in MP1 neurons

Measurement in naïve flies comparing control vs Dh44-R1 knockdown conditions did not reveal significant drop in pyruvate levels (Supplementary Figure 4B). It could be that, because of a basally increased ACC activity and fatty acid synthesis, feedback regulations maintain pyruvate at normal levels. But as explained above, the key point is that Dh44-R1 knockdown prevents the dynamic and experience-dependent regulation of pyruvate level that allows neuronal fueling after learning.

2. Support for double RNAi experiments

The authors should provide greater context for a series of experiments in which two RNAis, ACC and DH44, were expressed using drivers for two separate populations, cortex glia and MP1 neurons. We totally understand the technical impossibility of doing the ideal experiment (ACC RNAi only in cortex glia and DH44 RNAi only in MP1 neurons) and generally find this less-than-ideal method an acceptable compromise. However, to support and contextualize this series of experiments, answers to the following questions are important. Publicly available sequencing data should be able to clarify many of these.

We thank Reviewer #3 for acknowledging that our approach is perhaps the best currently achievable compromise. To bring further support that we indeed performed a separate functional knock-down of Dh44 in MP1 neurons and ACC in cortex glia, we comparatively investigated their expression levels and functional requirement in both cell types.

- Do neurons express ACC, since they do not form lipid droplets? Do they do so at levels comparable to glia?

We first sought to address this point at the transcriptomics level. Using the single-cell RNAseq dataset from Aso et al. (2018), we extracted the transcriptome of two neurons labeled as PPL1 γ 1-pedc (aka MP1 neurons). Both cells give quite consistent ranking of tyrosine hydroxylase (ple), Dh44, and ACC. In the first cell, out of ~9600 non-zero measured transcripts, TH ranks #12, Dh44 ranks #98 (~600 counts) and ACC ranks #5622 (ACC-RF :~3 counts). In the second one, out of ~18000 non-zero measured transcripts, TH ranks #136, Dh44 ranks #368 (~110 counts) and ACC ranks #6039 (ACC-RF :~5 counts). Thus, cell-wise, ACC expression is very low compared to Dh44, or to TH, another gene well-known to be functionally relevant for these neurons.

Unfortunately, this dataset does not include cortex glia for direct comparison of ACC levels with MP1 neurons within the same experimental conditions. On the other hand, it is not straightforward to identify unequivocally MP1 neurons in larger datasets such as the one from Davie et al., 2018. Therefore, to achieve a reliable comparison between MP1 neurons and cortex glia, we used two distinct datasets (Aso et al., 2018 ; Davie et al., 2018), using alpha/beta Kenyon cells as the common reference since they are present in the study by Aso et al., and are easily identifiable in the dataset published by Davie et al.

As in MP1 neurons, ACC-RF is the highly expressed ACC transcript in Kenyon cells. In the three alpha/beta Kenyon cells present in the Aso et al. dataset, ACC-RF counts are 17.24 ; 11.79 ; 22.68 respectively, i.e ~17 on average , more than 4 times higher than the ~4 counts measured in MP1 neurons.

Turning to the Davie et al. dataset, qualitative examination reveals expression scattered in many cell types, either glial or neuronal.

However, a more quantitative analysis of this dataset reveals on average a much higher transcript detection in cells of the cortex glia cluster than in the alpha/beta Kenyon cells cluster.

ACC transcripts (Davie et al. 2018)

In conclusion, the co-consideration of both datasets is supportive of the fact that ACC expression in MP1 neurons is much lower than in cortex glia.

To confirm this at the protein level, we conducted immunostainings in both cell types using a published and validated antibody against *Drosophila* ACC (Parvy et al., PloS Genetics, 2012). Cortex glia presented a strong cytosolic staining. In comparison, the cytosolic staining in MP1 neurons cell bodies was strikingly absent. Some staining was, unexpectedly, detected in the nucleus of MP1 neurons, but this is most likely not representative of ACC expression as (i) non-specific staining in the nuclear compartment was also reported in Parvy et al. ; (ii) in mammals, ACC localization was reported in the cytosol and the outer mitochondria membrane (Abu-Elheiga et al., 2000 ; Wang et al., 2022), as well as the endoplasmic reticulum membrane (Ivessa et al., 1997), but not inside the nucleus. Altogether, these stainings also point to a very low abundance of ACC in MP1 neurons. These stainings are presented in Supplementary Figure 1F.

Finally, at the functional and behavioral level, and consistent with these expression data, we would like to emphasize three important results (which were included in the initial version of the manuscript) that argue against a role of ACC in MP1 neurons in modulating memory either positively or negatively, or LD content in glia :

1°) Co-expression of Dh44 and ACC RNAi in MP1 neurons alone does not prevent the increase in LD in cortex glia (Fig. S1H); ACC in MP1 neurons is therefore not involved in the modulation of Dh44 release by these neurons.

2°) Likewise, co-expression of Dh44 and ACC RNAi in MP1 neurons does not prevent the memory defect induced by Dh44 knock-down (Supplementary Figures 5B and 6E).

3°) Expression of the single ACC RNAi in both cortex glia and MP1 neurons has no effect (no memory defect or increase) on middle- or long-term memory (this condition is included in the barplots of Figures 5C and 6D).

• Likewise, do cortex glia express DH44?

Here again, we addressed this question at the mRNA, protein and behavioral level, which all converge to the conclusion that Dh44 is very little (if at all) expressed in cortex glia, and not functionally involved in our behavioral task.

Using the Scope web interface to visualize the data from Davie et al. (2018), it is clear that brain cells expressing Dh44 and cells belonging to the cortex glia cluster do not overlap.

To confirm this at the protein level, we compared the level of Dh44 staining in MP1 neurons and cortex glia in the same brain. Taking as a common reference Dh44 signal in Pars Intercerebralis neurons, the staining level in MP1 neurons was much higher than in cortex glia (Data added on Supplementary Figure 1G).

Finally, we conducted memory assays on flies expressing the Dh44 RNAi in cortex glia only. We observed that neither middle- nor long-term memory was affected (Data added in Supplementary Figures 5C and 6E).

- If neuronal expression of ACC is low or absent, and/or cortex glia expression of Dh44 is low or absent, presenting this data (in a supplemental figure or table) could further strengthen confidence in Figure 1E, 4B, 4C, and 5D experiments that took the double RNAi (Dh44 and ACC simultaneous knockdown in cortex glia and MP1) approach. If this cannot be shown, we would suggest changing the figure headers from "Dh44 RNAi in MP1 and ACC RNAi in Cortex Glia" to "Dh44 and ACC RNAi in MP1 and Cortex Glia." Everywhere else the text is adequately clear.

Please note that former Figure 4B (now Figure 5B) was in fact not concerned by this remark, as in that case both Dh44-R1 and ACC RNAs were expressed in cortex glia only. For the other figures, although we agree that our approach is not ideal, we think the extended analysis of ACC and Dh44 expression that we now provide allows maintaining the figure headers as they were. The analysis of previously published transcriptomics dataset regarding ACC and Dh44 expression is provided as a Supplementary Note 1.

3. Conceptual framing

We are not necessarily convinced of the universal applicability of the idea that overall metabolic supply to the brain is constant, which appears both in the introduction (line 46) and the discussion (line 438). However, we also do not find this claim critical for the conclusions of this paper, and think it can be removed without much loss of impact. Previous work from this lab and others has shown flies eat more (Placais et al 2017) after certain types of memory training. Would this not suggest the brain is actively driving behaviors that increase the total energy available for memory consolidation, at least on the scale of hours? Even in the presence of increased energy supply, neurons could still receive that energy supply via glia, and glia could still be required to limit the amount they divert for fatty acid synthesis. Thus, there is no need to make this rather sweeping claim about constant total energy availability to the brain, and we find that other work undermines it in the case of *Drosophila* memory.

Our initial idea was to highlight evidence present in the literature to back up the notion that some energy-consuming processes could be downregulated to dampen the need of a global increase in energy uptake by the brain during memory formation. But we agree that claiming for a constant metabolic supply was probably going too far, and we removed the corresponding sentences in the introduction and at the end of the discussion.

Minor points

- The DH44 antibody was not properly validated in the Cabrero 2002 paper as far as we can see (as claimed on line 148), but we believe this manuscript has sufficiently validated it with the RNAi experiment in Figure 1C. If it were possible to mis-express Dh44 in a population that does not usually express it to serve as a positive control, that would be an encouraging further validation, but not absolutely required for the interpretation of experiments presented here.

We have corrected the sentence (previously line 148) to simply state that the antibody was previously published. As we could not find an existing, publicly available transgene to overexpress wild-type Dh44, and given that this was not an absolute requirement by Reviewer #3, we chose not to dig into further validation of the antibody.

- Methods - Please provide the exact recipe for the fly food used.

This has been done (line 483).

- Methods - Please clarify memory experiment methods: was there any set threshold for the number of flies making a choice for the data point to be counted? For example, if one fly chose one odor, one fly chose the other odor, and 28 flies remained in the center, would this data point be discarded, or would it be included as a PI of 0?

As a general rule, we discard memory scores that involve less than 6 flies, to avoid giving too much statistical importance to the choice of a single fly. This is now specified in the Methods (line 519).

- Methods – Please specify the time of day memory experiments were performed (See: <https://www.biorxiv.org/content/10.1101/2024.01.25.577231v1>)

Experiments were performed in the morning or in the afternoon indiscriminately. Every single graph of memory experiment usually includes data collected at different timepoints. We did

not notice qualitative differences in the results among experiments performed in the morning or in the afternoon. We added a sentence to mention that data were collected anytime during the light phase of the flies (line 496).

- Line 56: “fruit flies” (two words)
- Line 88: “memorize” should probably just be “remember” (“memorize” has connotations of deliberately studying something)
- Line 125: “confirmed” should be “resulted in” or “caused” or “produced” or something like that
- Line 141: it would be clearer to write “PI neurons” instead of “Dh44 neurons” since it will be soon revealed that there are more Dh44 neurons than previously thought

We have done the suggested corrections.

Reviewer #4 (Remarks to the Author):

I co-reviewed this manuscript with one of the reviewers who provided the listed reports. This is part of the Nature Communications initiative to facilitate training in peer review and to provide appropriate recognition for Early Career Researchers who co-review manuscripts